

# Large-scale coastal and fluvial models constrain the late Holocene evolution of the Ebro delta, Spain

Jaap H. Nienhuis[1,2], Andrew D. Ashton[2], Albert J. Kettner[3], Liviu Giosan[2]

5  [1] Earth, Atmospheric and Planetary Science, Massachusetts Institute of Technology, Cambridge, MA 02139
[2] Geology and Geophysics, Woods Hole Oceanographic Institution, Woods Hole, MA 02543
[3] Institute of Arctic and Alpine Research, University of Colorado, Boulder, CO 80309

*Correspondence to*: Jaap H. Nienhuis (jnienhui@tulane.edu)



**Abstract.** The distinctive plan-view shape of the Ebro Delta, Spain, reveals a rich morphologic history. The degree to which the form and depositional history of the Ebro and many other deltas represent autogenic (internal) dynamics or allogenic (external) forcing remains a prominent challenge for paleo-environmental reconstructions. Here we use simple coastal and fluvial morphodynamic models to quantify paleo-environmental changes that affected the Ebro delta over the late Holocene. Based on

5    numerical model experiments and the preserved and modern Ebro delta shape, we estimate that a phase of rapid shoreline progradation began approximately 2100 years BP, requiring a large increase (doubling) in coarse-grained fluvial sediment supply to the delta. We do not find evidence that changes in wave climate aided this delta expansion. River profile models suggest that such an instantaneous and sustained increase in coarse-grained, beach-compatible sediment to the delta would require a combination of flood discharge increase and increased sediment input into the river channel from upstream drainage basin erosion.

10   The persistence of rapid delta progradation throughout the last 2100 years suggests an anthropogenic signal of sediment supply and flooding intensity. Our findings highlight how scenario-based investigations of deltaic systems using simple models can assist first-order quantitative paleo-environmental reconstructions, elucidating the effects of past human influence and climate change and allowing a better understanding of the future of deltaic landforms.



## 1        Introduction

The Ebro Delta, Spain, with its distinctive plan-view shape, has experienced significant morphologic changes over the last millennia caused by the growth and reworking of different delta lobes (Fig. 1) (Canicio and Ibáñez, 1999). While autogenic delta processes might have caused some of these morphological changes, others aspects could be attributable to past climate changes or
anthropogenic activities within the drainage basin. Many different scenarios leading to the modern morphology have been proposed, including high-frequency (centennial scale) sea level fluctuations (Somoza et al., 1998), human-induced sediment load changes in the Ebro River (Guillén and Palanques, 1997a), and climate fluctuations affecting river discharge (Xing et al., 2014).

Many deltas around the world have experienced substantial morphologic changes over the last millennia due to anthropogenic
factors such as river damming, land-use change, and climate change (Anthony et al., 2014; Giosan et al., 2012; Maselli and Trincardi, 2013; Syvitski and Saito, 2007). The Ebro delta lends itself particularly well to quantitative reconstructions because it is morphologically constrained (Nelson, 1990); it displays a distinctive plan-view shape (Fig. 1); and its environment is relatively well-studied (Cearreta et al., 2016; Maldonado, 1975). Here, we use a coastline evolution model and a river profile evolution model to quantitatively constrain the style, timing, and rate of Ebro delta morphologic change and the associated fluvial transport
conditions towards the Ebro delta during the Holocene.

Our goal in this paper is to investigate the general evolution of the Ebro delta-river system using "scenario-based" and quantitative model experiments. We do not attempt to capture the precise morphology or geochronology of any one segment of the Ebro delta, but rather approximate delta paleo-morphodynamics to assess the potential physical mechanisms that could have formed this delta
plain. Our scenario-based approach allows us to test existing hypotheses of environmental changes that may have affected the Ebro delta's development, and to quantify first-order sediment fluxes and timescales. As a test of the suitability of the delta and the river models, we compare the model predictions to observed deltaic (Cearreta et al., 2016; Jiménez and Sánchez-Arcilla, 1993) and fluvial change (Vericat and Batalla, 2006) over the last century.

## 2        Background

### 25    2.1        Ebro River

The Ebro River reached the Mediterranean Sea, after an endorheic phase, sometime between 13 and 5 million years ago (Babault et al., 2006; Garcia-Castellanos et al., 2003). Its modern drainage basin extends over 85,530 km$^2$, covering a large portion of the Pyrenees, the Cantabrian mountains, and the Iberian massif (Mikeš, 2010). The average channel width in the lower course of the river is ~150 m, with a bankfull flow depth of ~5 m (Guillén and Palanques, 1997a). Average (pre-dam) discharge has been
estimated at about 500 m$^3$s$^{-1}$ (Batalla et al., 2004). The fluvial sediment flux during the Holocene highstand, based on radiometric dating of Ebro continental shelf deposits, is estimated to be ca. 200 kg s$^{-1}$ (6.3 MT yr$^{-1}$) (Nelson, 1990). The suspended load consists mostly of clay and silt (Muñoz and Prat, 1989), while the bedload is predominantly sand and gravel (Vericat and Batalla, 2006).

### 2.1        Ebro Delta

At the Ebro River outlet to the Mediterranean Sea, fluvial sediment deposition over the course of millions of years has expanded
the Ebro continental shelf and constructed successive deltas (Babault et al., 2006; Nelson, 1990). During the Holocene, strong waves and limited coarse-grained sediment input have shaped the Ebro coast towards a wave-dominated delta with a smooth shoreline and single thread distributary network (Jiménez et al., 1997). The Ebro nearshore zone consists mostly of sand size



sediment (Maldonado, 1975; Somoza et al., 1998) to a depth of ~12 m, transitioning into muds farther offshore (Guillén and Palanques, 1997b). Two distinctive features on the Ebro delta are the spits to the north (El Fangar) and south (La Banya) of the current river mouth, considered to be formed by wave reworking of abandoned delta lobes (Fig. 1) (Maldonado, 1975).

### 2.3 Ebro Delta Holocene evolution

Similar to many other deltas around the world, Holocene sea level rise led to the transgression of the last Pleistocene Ebro delta (Maldonado, 1975). The maximum flooding surface of the Ebro delta is dated to about 6900 years BP, with its landward extent near the town of Amposta (Lario et al., 1995; Somoza et al., 1998). Several studies have interpreted historical references to suggest that the Ebro was still an estuary ~2000 years ago (Guillén and Palanques, 1997a; Maselli and Trincardi, 2013); however, radiocarbon dating of relict beach ridges (Canicio and Ibáñez, 1999) and recently dated sandy beach shells from boreholes indicate

that the delta was already formed by ~6000 years BP (Cearreta et al., 2016).

These dated beach ridges show that the Ebro delta plain was small, cuspate, and wave-dominated at least until 3000 years BP (Canicio and Ibáñez, 1999). The same study suggested that sometime between 1400 and 1000 years BP the Riet Vell lobe had grown rapidly and extended approximately 20 km into the Mediterranean Sea, although no confirming dates are currently available.

This increase in progradation rate, at least 2 to 3 times faster than previous and initiating sometime after 3000 but before 1400 years BP, is commonly ascribed to land use changes and/or climatic variability increasing fluvial sediment supply (Thorndycraft and Benito, 2006).

What could have caused this increase in fluvial sediment supply? Benito et al. (2008), dating floodplain alluviation of Spanish

rivers, suggested three periods of intense flooding over the last three millennia: 2710-2320 years BP, 2000-1830 years BP, and 910-500 years BP. The first of these three periods has been associated with large-scale climate variability causing increased flooding. The last period of floodplain aggradation however is not in phase with palaeoflood records, which Benito et al. (2008) therefore attributed to anthropogenic modifications such as deforestation that increased the Ebro River sediment load. Other deltas around the Mediterranean and the Black Sea, whose hinterlands have comparable observed land-use change histories, also show

periods of increased progradation in response to human activities (Anthony et al., 2014; Giosan et al., 2012; Maselli and Trincardi, 2013).

Xing et al. (2014) used the long-term fluvial discharge and sediment supply model HydroTrend (Kettner and Syvitski, 2008) to quantify the effects of anthropogenic and climate change on fluvial suspended sediment supply to the Ebro delta. HydroTrend uses

empirical relations between, among others, basin area, land-cover, drainage basin relief, temperature, and precipitation and is calibrated using modern sediment transport records to simulate river sediment load. The model results of Xing et al. (2014) suggest that discharge variation was mostly a result of climatic variability, whereas forest clearing likely contributed to changes in suspended sediment load. Their study estimated a 40% increase in the fluvial suspended sediment load in response to deforestation. Other studies, such as Nelson (1990) and Guillén and Palanques (1997a), who reconstructed a sediment budget from delta plain

and shelf aggradation rates, have estimated a greater fluvial sediment flux increase of 350%.

Relict channel deposits on the delta plain (Maldonado, 1975) combined with published maps and historical evidence (Canicio and Ibáñez, 1999; Somoza and Rodriguez-Santalla, 2014) suggest that the progradation of the Riet Vell lobe stopped after 1000 years BP but prior to 600 years BP, when the avulsion of the main channel started the new Sol de Riu lobe to the north (Fig. 1).



Subsequently, the Riet Vell lobe was reworked into the La Banya spit to the south. After a second river avulsion about 300 years ago to the Mitjorn-Buda lobe in between the previous active channels, the Sol de Riu lobe was also abandoned and reworked into the northern El Fangar spit (Fig. 1).

### 2.4    Recent changes

Starting in the 20[th] century, over 187 dams have been built in the Ebro that have highly regulated its discharge and currently impound 57% of the mean annual runoff (Batalla et al., 2004). The average fluvial water discharge based on hydrographic records before dam construction was approximately 500 $m^3 s^{-1}$, while post-dam discharge has averaged about 340 $m^3 s^{-1}$ (Batalla et al., 2004).

Prior to the construction of the major dams in the Ebro, peak discharge was about 50% higher than current values (Batalla et al., 2004). As a consequence, while bedload-transporting river flows (>860 $m^3 s^{-1}$) were previously exceeded 15% of the time, dams reduced the exceedance frequency of these floods to just 4% of the year and thereby lowered the bedload sediment flux at the delta outlet (Vericat and Batalla, 2006). Additionally, reservoirs behind dams trap about 90% of the upstream suspended sediment load and 100% of the upstream bedload (Vericat and Batalla, 2006).

Accurate measures of pre-dam fluvial sediment flux are challenging, but from early 20[th] century sediment concentration and discharge measurements, Guillén and Palanques (1992) obtained an annual average suspended load estimate of ~600 kg $s^{-1}$ (20 MT $yr^{-1}$). Syvitski and Saito (2007) used Bagnold's (1966) equation to estimate a pre-dam bedload flux of 71 kg $s^{-1}$ (2.2 MT $yr^{-1}$).

Post-dam measurements taken 50 km upstream of the delta (25 km downstream of the Flix dam, the last major dam in the main river channel) estimate a modern total sediment load of about 28 kg $s^{-1}$ (0.9 MT $yr^{-1}$), of which 40% is transported as bedload (Vericat and Batalla, 2006). Using predictive sediment transport formulae from van Rijn (1984) combined with discharge measurements, Jiménez (1990) estimated the modern sand (bedload) transport at the mouth of the delta at 1.6 kg $s^{-1}$ (0.05 MT $yr^{-1}$).

A comparison of estimates of pre-dam to post-dam bedload transport to the delta suggests a reduction of about ~ 95%. Evidence of this sediment deficit include scours of the lower course of the channel bed and the formation of armored layers. Immediately downstream of the Flix dam, the channel bed surface consists of coarse gravel ($D_{50}$ = 38mm) while the subsurface consists of mixed sand and gravel ($D_{50}$ = 17mm) (Vericat et al., 2006).

The 20[th] century fluvial sediment flux reduction has also led to morphologic changes of the delta at the coast. While for much of the last millennia, the Ebro delta mouth was probably, at least periodically, close to a river-dominated morphology, the sediment supply reduction has led to a more wave-dominated form of the modern Mitjorn-Buda lobe (Jiménez et al., 1997).

River damming may not be the only cause for large-scale coastal changes in the future. A bath-tub-style estimate projected that

subsidence and sea level rise may submerge about 40% of the delta surface by 2100 (Ibáñez et al., 1997). However, the projected effects of sea-level rise on coastal change up to 2050 are negligible compared to ongoing change resulting from alongshore sediment transport gradients (Sánchez-Arcilla et al., 2008). These gradients have caused retreat rates of 50 m $yr^{-1}$ near the river mouth, and have resulted in spit accretion at rates of approximately 10 m $yr^{-1}$ between 1957 and 1992 (Jiménez and Sánchez-Arcilla, 1993).





### 2.5 Modeling wave-influenced deltas

Many numerical models have been developed over the last decades to quantitatively reproduce, predict, and understand the dynamics of deltaic systems. Complex 'simulation models' such as Delft3D typically are used to reproduce a particular well-constrained natural environment (e.g., van der Wegen et al., 2011) or to parameterize poorly understood physical processes (e.g.,

Nienhuis et al., 2016a). Simple 'exploratory models' of 'reduced complexity' on the other hand are designed to capture the essential feedbacks leading to an observed phenomenon (Murray, 2003). Because, on the long-term, the millennial- to centennial-scale development of the Ebro delta is poorly constrained, here, we apply exploratory models of wave-influenced delta dynamics to capture the essential physical mechanisms affecting the evolving morphology of the Ebro delta using scenario-based approaches.

The plan-view shape of the Ebro delta, like other wave-dominated deltas, is governed by wave-driven alongshore sediment transport (Bakker and Edelman, 1964; Bhattacharya and Giosan, 2003; Jiménez and Sánchez-Arcilla, 1993). Modeling of wave-dominated delta shape is therefore usually performed with coastline models (e.g., Ashton and Giosan, 2011; Bakker and Edelman, 1964). By assuming the cross-shore profile maintains a constant shape, gradients in alongshore transport can be linearly related to accretion or erosion any one contour line, typically the coastline. Such one-contour-line models calculate alongshore sediment

transport based on surf-zone averaged equations such as the CERC formula (Komar, 1971), which relate the relative wave angle and height to a sediment transport flux. The cuspate coastline shape typical of wave-influenced deltas arises when adding a point-source of (fluvial) sediment to an otherwise straight sandy coast (Grijm, 1960).

By comparing fluvial and wave-driven sediment fluxes, Nienhuis et al. (2015) quantified when deltas would be expected attain a

wave-dominated versus a river-dominated shape. If the fluvial sediment supply is larger than the maximum potential alongshore sediment transport away from both delta flanks, waves cannot transport fluvial sediment delivered at the river mouth alongshore and a delta would be expected to be river-dominated. Their study defined a river dominance ratio $R$ as the fluvial sediment flux ($Q_r$) divided by the maximum alongshore sediment transport flux away from the river in both directions ($Q_{s,max}$). For $R > 1$, the delta is river-dominated. If $R < 1$, there is an equilibrium plan view delta flank orientation such that the fluvial sediment flux ($Q_r$)

equals the wave-driven sediment flux ($Q_s$) away from the river mouth along both flanks. The amount of shoreline deflection at the river mouth of a wave-dominated delta is therefore an indicator of its wave dominance.

Ashton and Giosan (2011) showed that for very obliquely approaching waves, wave-dominated deltas can become asymmetrical and develop downdrift migrating sand waves and spits on the downdrift flank. These shoreline instabilities can form on growing

deltas in which case they are oriented roughly parallel to the delta flank. Nienhuis et al. (2013) later showed that prominent recurved spits can develop from the reworking of delta lobes. These recurved spits develop after a reduction in fluvial sediment supply to a delta lobe (e.g. due to avulsion or dam construction) only if one or both flanks of the delta grew past the maximum in alongshore sediment transport. These recurved spits are generally not oriented parallel to the delta coastline. Rather, the orientation of free spits is controlled by the wave climate and the rate of delta lobe retreat (Ashton et al., 2016; Nienhuis et al., 2013).

### 2.6 Modeling fluvial sediment supply

The sand-sized sediment feeding the Ebro delta is supplied as bedload and suspended load through the Ebro river, interacting with the alluvial river bed (Jiménez et al., 1990). In alluvial rivers, channel-bed interaction sets up an equilibrium between the along-stream slope, river discharge, and sediment supply (Lane, 1955). One of the first attempts to numerically model fluvial sediment transport was by Hirano (1971), who combined the depth-averaged, one-dimensional Saint-Venant equations for fluid flow with a





simple formulation for sediment transport. Their model resulted in a typical concave up longitudinal river profile for a scenario of gradually increasing water discharge downstream (Hirano, 1971; Snow and Slingerland, 1987).

For normal flow conditions, the Saint-Venant equations can be simplified substantially by formulating an alongstream momentum

balance that relates bed shear stress to water depth and bed slope. River profile models are usually combined with an Exner equation for sediment conservation and a Chezy or Manning coefficient for form drag. For normal flow conditions this combination results in a simple analytical expression for longitudinal river profile shape and equilibrium sediment transport rates (Parker, 1978).

The normal flow assumption breaks down if the flow is sufficiently non-steady, such as in backwater zones in the vicinity of a

river delta (Hotchkiss and Parker, 1991). In that case, sediment is deposited in the backwater zone upstream of the river mouth but also in the delta foreset downstream of the river mouth. Even though the normal flow assumption is no longer valid in the backwater zone, generally sediment deposition during low flow nearly balances erosion during fluvial floods (Chatanantavet et al., 2012) such that the foreset can be considered the dominant location of bedload sediment deposition. In our simplified Ebro delta river profile model, we therefore assume that all bedload sediment transported to the apex of the Ebro delta is deposited near the river

mouth as delta foreset.

### 3 Methods

#### 3.1 Delta evolution model

We study the morphologic evolution of the Ebro delta using the Coastline Evolution Model (CEM), an exploratory, process-based one-contour-line model (for a full description see Ashton and Murray, 2006). In this model, the plan-view coastal zone is

discretized into 50 m square cells that are either filled (land), empty (water), or partially filled (coastline), the latter allowing for a smooth, continuous shoreline. Incoming deep-water waves are refracted and shoaled from the toe of the shoreface up to the breaking wave depth assuming parallel shoreline contours. We calculate alongshore sediment transport $Q_s$ (kg s$^{-1}$) with the CERC formula (Komar, 1971), using the wave height and the relative wave approach angle to determine the sediment flux across different shoreline cells:

$$Q_s = K_1 \cdot \rho_s \cdot (1-p) \cdot H_s^{12/5} T^{1/5} \cos^{6/5}(\phi_0 - \phi_s) \sin(\phi_0 - \phi_s), \qquad (1)$$

where $H_s$ is the offshore deep-water significant wave height (m), $T$ is the wave period (s), $\phi_0$ is the deep-water wave approach angle (which equals $\gamma - \theta$ in a regional setting, Fig. 3a), and $\phi_s$ is the local shoreline orientation (Ashton and Murray, 2006; Nienhuis et al., 2015). The density of sediment is $\rho_s$ (kg m$^{-3}$) and $p$ is the dry mass void fraction. From Ebro delta calibration studies of Jiménez and Sánchez-Arcilla (1993), we use a littoral transport coefficient $K_1$ of 0.035 m$^{3/5}$ s$^{-6/5}$ compared to the typical coefficient of 0.06

m$^{3/5}$ s$^{-6/5}$ (Komar, 1998).

Following the one-contour-line approach, the divergence of alongshore sediment transport is related to shoreline accretion or erosion up to the shoreface depth using the shoreline Exner equation,

$$\frac{d\eta}{dt} = -\frac{1}{D_{sf}} \frac{1}{(1-p) \cdot \rho_s} \frac{dQ_s}{dx}, \qquad (2)$$

where $d\eta/dt$ is erosion or progradation of the shoreline (m s$^{-1}$), $D_{sf}$ is the shoreface depth (m), and $dQ_s/dx$ is the alongshore gradient in alongshore sediment transport (kg s$^{-1}$ m$^{-1}$).

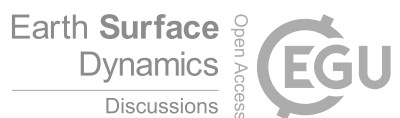

An advantage of the CEM is its ability to produce arbitrarily sinuous shoreline shapes such as spits. When shoreline erosion causes a neck of a spit to reach a critical width, overwash occurs and sediment is transported from the shoreface to the backbarrier to maintain a minimum width (Jiménez and Sánchez-Arcilla, 2004). Overwash allows spits and barriers to retreat without

disconnecting from the rest of the coastline (Ashton and Murray, 2006). Following observations of Jiménez and Sánchez-Arcilla (2004) of the La Banya spit, we set the critical barrier width to 250 m. The overwash depth is determined geometrically assuming a shoreface slope (0.01) and an overwash volume. Even though this is obviously a simplification that could result in overwash depths that are unrealistically deep, it avoids the need for a complicated assessment of backbarrier elevations coastwide.

From bed-surface samples of the Ebro delta coastline, Guillén and Palanques (1997b) found that the sand-mud transition is located at approximately 12 m water depth. In a study of short-term (decadal) coastal change, Jiménez and Sánchez-Arcilla (1993) suggest a 7 m depth of closure based on cross-shore profile variability. In our model, we choose an intermediate shoreface depth of 10 m. The characteristic shoreface slope (0.01) and shelf slope (0.002) are set based on the geometry of the Ebro Delta (Guillén and Jiménez, 1995; Jiménez and Sánchez-Arcilla, 1993).

In CEM, the river channel is highly simplified and is only represented as the location alongshore where the littoral-grade portion of the fluvial sediment is deposited. By modeling the mass balance this way, we assume that fine-grained fluvial sediment is winnowed by waves and eventually deposited largely offshore beyond the shoreface (Guillén and Palanques, 1997b). As the delta progrades or retreats, the channel location follows a predefined trajectory from the river apex based on observed Ebro delta (paleo)

channel trajectories of the Riet Vell, Sol de Riu, and Mitjorn-Buda lobes (Fig. 2).

### 3.2 Application to the Ebro Delta

We have adapted CEM to model growth and reworking of the different Ebro delta lobes. Rather than growing perpendicularly to the initial coastline, we force individual channels to grow along channel paths that we choose based upon the paleo and modern channels of the Ebro delta (Fig. 2, Maldonado, 1975). The first lobe builds out at 5º from shore normal and represents the growth

of the Riet Vell lobe. The second (Sol de Riu) lobe grows -45º from shore normal, and the modern Mitjorn-Buda lobe is oriented at -20º. As a second modification to the original model, we disable alongshore sediment transport out of a cell that is part of the initial coastline. This modification accounts for the fact that the Ebro Delta juts out of the rocky coastline of Mediterranean Spain, and is not connected to an updrift littoral sediment source (Fig. 2).

Even though the Ebro delta channel orientations are likely in part determined by wave climate, fluvial sediment supply, and alongshore sediment bypassing of the river mouth (i.e., Nienhuis et al., 2016b), we choose to impose channel orientations directly to contain model variability. Similarly, delta avulsion has been suggested to be controlled by backwater length and channel filling time scales (Chatanantavet et al., 2012). To limit model sensitivity we do not allow autogenic river avulsions in our model, instead we model avulsions at their historically inferred locations (Maldonado, 1975) at which we impose avulsion times directly.


It is important to note that we are not explicitly simulating the history of the Ebro River delta; rather we use simple models to constrain fluvial sediment fluxes and delta growth in a broadly representative wave-dominated environment. We run scenarios of different fluvial sediment supply rates to investigate its effect on Ebro delta morphology, including the characteristic spits. We





also run scenarios of different channel avulsion timings and match the resulting modeled delta shape to the modern Ebro delta shape to constrain Ebro delta geochronology.

### 3.3 Wave climate

Wave height and the directional distribution of incoming waves exert a first-order control on wave-influenced delta evolution
(Ashton and Giosan, 2011). We compared five different wave climatology sources from nearby the Ebro delta and investigated their effect on modeled alongshore sediment transport. Wave climates are extracted from two directional wave buoys and three hind-casted wave models (Fig. 3 and Table 1). All sources are located in sufficiently deep water for the waves to be treated as deep-water waves (depth $> ¼ \; \pi \, T_P^2$), and all sources show peaks of wave intensity from the East and from the South that affect Ebro delta alongshore transport. The different wave sources differ particularly in the relative strength of the waves approaching
from the south. This could be because the southerly (summer) waves are generated more locally (Jiménez et al., 1997) and therefore their magnitude may be sensitive to buoy location or hind-cast methodology.

### 3.4 Testing the alongshore sediment transport model assumptions

Jiménez and Sánchez-Arcilla (1993) used aerial photographs from 1957 to 1989 and beach profile measurements between 1988 and 1992 to calculate Ebro coastline change. Their study found sustained multi-decadal rates of erosion of up to 50 m yr$^{-1}$ close to
the river mouth, and progradation of about 10 m yr$^{-1}$ along the spits (Jiménez and Sánchez-Arcilla, 1993). These measured recent shoreline changes allow us to test the one-line shoreline assumptions underlying the delta evolution model. With the CERC formula (eq. 1, Komar, 1971), and using the 5 wave sources (Table 1 and Fig. 3), we computed net alongshore sediment transport along the modern Ebro delta shoreline extracted from the NOAA shoreline database (NOAA, 2015), taking into account shadowing of certain wave approach angles by other portions of the delta coastline.

The calculated littoral sediment transport trends along the Ebro delta coastline are similar between the five wave climates (Fig. 4), showing sediment transport is greatest along both spits and close to the modern river mouth. The computed sediment transport magnitude however between the wave climate sources differs by almost a factor of 3. All wave climates except for the MedAtlas have similar correlation coefficients when compared to sediment transport patterns estimated based on observed beach change
(Fig. 4b, black markers) (Jiménez and Sánchez-Arcilla, 1993). We choose to use the Cap Tortosa buoy data (described in Bolanos et al., 2009) in the delta evolution model because its 21 year record is sufficiently long, it is located close to the mouth of the modern Ebro river, and its wave height and wave period are average compared to the other 4 wave sources.

From the computed alongshore sediment transport gradients from the Cap Tortosa data, we predict shoreline accretion and erosion
using the one-contour-line approach and the same shoreface depth and littoral transport constant as the delta evolution model. In general, the rate of shoreline change is well predicted ($R^2 = 0.84$) by the one-contour-line model and the wave climate from the Cap Tortosa buoy (Fig. 4c).

Aside from testing our model, we can draw two observations from the measurements of Jiménez and Sánchez-Arcilla (1993) about
the ongoing coastal changes of the Ebro delta. First, around the river mouth there is rapid coastal retreat to the south, and deposition further to the north. The field measurements align with the one-contour-line predictions close to the river mouth without including a fluvial sediment contribution, which provides further evidence of the negligible fluvial sediment supply to the coast (Jiménez and Sánchez-Arcilla, 1993).





Secondly, the sediment transport patterns along the spits can be cast in the framework proposed by Ashton et al. (2016). Along the barrier sections of the Ebro Delta spits, the computed alongshore sediment transport gradients are nearly zero, whereas measured shoreline retreat is approximately 10 m yr⁻¹ (Fig. 4c). This suggests that overwash is driving coastline retreat without gradients in

alongshore sediment transport. The barrier section (the "neck") is fed by a sediment source upcoast and is generally erosional up to a fulcrum point, where alongshore sediment transport is maximized and erosion transitions into deposition (Ashton et al., 2016). The measured and predicted shoreline change indicate that the northern and the southern spit are indeed depositional and are prograding at approximately 10 m yr⁻¹ (Fig. 4c).

### 3.5    River modeling

We investigate the response timescales of the river basin to climate and land-use changes using an exploratory 1-D river profile model (Parker, 2004). In this model, sediment is not merely a passive tracer, but interacts with the bed elevation to reach a longitudinal profile in morphodynamic equilibrium (Carling and Cao, 2002). The interaction between flow and topography creates a dynamic model – rivers are not treated as static pipes – which allows us to use the computed longitudinal profiles together with the observed modern longitudinal profile to investigate potential past and present sediment transport conditions. Additionally, by

focusing on the interaction of the flow with the channel bed, we can model the bed material load – the sediment that makes up most of the delta (Maldonado, 1975) – while we ignore the finer grained material that is mostly deposited farther offshore. The channel bed in the model is freely erodible and our approach is therefore strictly applicable to alluvial, transport-limited systems (Parker, 2004). A similar 1-D river profile model was recently applied to study timescales of sediment supply decreases in the Mississippi River (Nittrouer and Viparelli, 2014). Their study suggested a long ($O$ 100 yr) delay between dam construction ~1000

km upstream and sand load changes near the coast.

The 1-D river profile model assumes normal flow conditions, such that a width-averaged momentum balance connects bed slope and flow depth to bed shear stress. Flow in the channel is determined using a Manning-Strickler formulation for the flow resistance (Parker, 2004). The model uses the Meyer-Peter and Muller (1948) equation to calculate fluvial sediment transport (kg s⁻¹),

$$Q_r = I\rho_s B\sqrt{RgD}D\alpha_t \left\{ \left[ \left( \frac{Q_{flood}^2 k_c^{1/3}}{\alpha_r^2 gB^2} \right)^{3/10} \frac{S^{7/10}}{RD} \right] - \tau_c \right\}^{n_t}, \tag{3}$$

where $R$ is the submerged specific gravity of the sediment (1.65); $g$ is gravity (m s⁻²); $D$ is the median grain size (m) which we choose to be the littoral grain size of 0.2 mm (Jiménez and Sánchez-Arcilla, 1993); $\alpha_t$, $\alpha_r$, and $n_t$ are sediment transport coefficients; $Q_{flood}$ is the flood discharge (m³ s⁻¹); $k_c$ is the bed roughness (m); $S$ is the channel bed slope; $I$ is the flood discharge intermittency; $\rho_s$ is the sediment density (kg m⁻³); $B$ is the channel width (m); and $\tau_c$ is the non-dimensional critical bed shear stress for sediment

motion (0.0495) (Parker, 2004).

The normal flow assumption is invalid in the backwater zone near the delta, where the channel aggrades and prograues (Hotchkiss and Parker, 1991). Technically therefore, the apex of the delta should be considered the downstream boundary of the fluvial profile model. However, as Chatanantavet et al. (2012) recently showed, annual flooding cycles in the backwater zone often create a

condition where aggradation during low flow is nearly balanced by erosion during high flow. This (near) balance suggests that in terms of bedload volumes, delta progradation is significantly larger than channel aggradation and, therefore, that the absence of a



backwater zone in our normal flow model only results in a limited underestimation of the fluvial sediment supply to the river mouth when considering centennial timescales.

Following Jiménez and Sánchez-Arcilla (1993), we choose a grain size of 0.2 mm for the fluvial profile model. This grain size is
mostly transported during floods of 900 m³ s⁻¹ or larger; flows which during pre-dam conditions were exceeded 15% of the time (Batalla et al., 2004). The fluvial profile model uses one flood magnitude with an intermittency factor rather than an exceedance frequency. We have estimated an intermittency factor by fitting and integrating a logarithmic trend to the flood frequency analysis data of Batalla et al. (2004). This integration shows that the sediment load of a 900 m³ s⁻¹, 15% *exceedance* frequency flow roughly corresponds to a 900 m³ s⁻¹, 30% *intermittency* factor flow, which we therefore use in the model.

We can compare the predictions from the model to the observed modern river profile and see how close the modern profile is to equilibrium. The modern Ebro River profile (Fig. 5) shows an approximately constant slope up to the confluence with the Arga River, 450 km upstream. Applying the model based on the pre-dam fluvial and discharge conditions ($D_{50}$ = 0.2 mm, $Q_{flood}$ = 900 m³ s⁻¹, I = 30%, $Q_r$ = 70 kg s⁻¹), we find that the equilibrium slope is estimated surprisingly well (5.8·10⁻⁴, Fig. 5b). Note that the
observed channel slope remains constant upstream of the confluence with Cinca River even though the flood discharge decreases significantly. This could be due to different channel bed grain sizes between the Cinca River and the Ebro River upstream of this confluence. We model the Ebro drainage basin as a single channel representing an average of its tributaries. A spatially explicit model of the Ebro basin would be a significant departure from our exploratory model approach.

The 1-D river profile model requires the choice of an upstream boundary, representing the average location of the fluvial discharge and sediment supply into in the drainage basin. The choice of an upstream boundary is important because it acts as a first-order control on fluvial sediment transport timescales from the basin to the delta. To find an appropriate upstream boundary, we calculated the pre-dam morphologic (2-year) flood discharge along the Ebro river relative to the discharge at the delta from existing hydrologic records (Batalla et al., 2004). We set the upstream boundary condition at 450 km upstream of the delta, where the Ebro
river pre-dam morphologic (2-year) flood discharge is 50% of its final discharge at the delta and a clear discontinuity in the longitudinal profile occurs (Fig. 5).

### 3.6    Testing the fluvial profile model

To test the applicability of the river profile model to the Ebro drainage basin, we compare model estimates to recent measured bed elevation and sediment transport changes 25 km downstream of the lowermost Flix Dam for 55 years after its construction in 1948
(Fig. 6) (Vericat and Batalla, 2006). Between 2002 and 2004, Vericat and Batalla (2006) observed an average bedload transport rate of 12 kg s⁻¹, down from pre-dam estimates of around 70 kg s⁻¹. They also observed downstream scour at a rate of about 0.03 m yr⁻¹ in Mora d'Ebre (Fig. 6b). To model river profile response to dam construction, we applied a 100% reduction in sediment supply immediately downstream of the Flix Dam. Concomitantly, following analysis of Vericat and Batalla (2006), we impose a fourfold decrease in the occurrence of bedload transporting floods of 900 m³ s⁻¹ (from a 15% to a 4% exceedance probability, or a
30% to an 8% intermittency factor).

Even though the model does not capture processes such as bed armouring and downstream fining, results show reasonable agreement with the field measurements, estimating about 1 m of bed degradation at Mora d'Ebre 50 years after dam construction (0.02 m yr⁻¹), and a local sediment bedload of 16 kg s⁻¹. Furthermore, the modeled bed response to dam construction has not yet




reached the Ebro delta. At Mora d'Ebre, the measurement location of Vericat and Batalla (2006), equation (3) predicts that the change in flooding frequency decreased the coarse grained sediment flux from 70 kg s$^{-1}$ to 19 kg s$^{-1}$. The sediment capture in the reservoirs and the subsequent channel bed slope adjustment decreased the coarse grained sediment flux further from 19 kg s$^{-1}$ to 16 kg s$^{-1}$. Therefore, of the total reduction, the model predicts that about 95% is due to changes in the flooding frequency, whereas

only 5% is due to a capturing of the sediment in the reservoirs.

## 4    Results

### 4.1    Delta response to increased fluvial sediment supply

We investigated if changes in fluvial sediment supply could explain the rapid growth of the Riet Vell lobe that is thought to have occurred sometime between 3000 and 1100 years BP (Canicio and Ibáñez, 1999). Cast in terms of the fluvial dominance ratio $R$,

which equals $Q_r / Q_{s,max}$ (Nienhuis et al., 2015), the transition from a slowly growing cuspate delta to a rapidly growing pointy (not cuspate) delta occurs when $R > 1$ (or $Q_r \sim 50$ kg s$^{-1}$, see Table 1). At a pre-dam estimate of 70 kg s$^{-1}$ (Syvitski and Saito, 2007), this means that during the period of rapid growth, the Ebro delta should have been river-dominated or close to a transition to river dominance, with a fluvial dominance ratio $R$ of 1.4.

We also investigated the effect of fluvial sediment supply on Ebro Delta morphology with the delta evolution model. After 750 model years, for bedload sediment fluxes up to about 35 kg s$^{-1}$ (1 MT yr$^{-1}$), the modeled delta exhibits a smooth cuspate morphology (Fig. 7a) while prograding at about 6 m yr$^{-1}$ (5 km in 800 years, Fig. 7c). A delta supplied with a sand load of 70 kg s$^{-1}$ (2 MT yr$^{-1}$), however, progrades more rapidly at ~30 m yr$^{-1}$ and forms shoreline instabilities along the updrift and downdrift flanks.

From the same set of model experiments, we can also study the effect of fluvial sediment supply on post-avulsion abandonment and wave reworking. For low pre-abandonment fluvial sediment supply (< 40 kg s$^{-1}$), the delta remains wave-dominated during growth ($R < 1$) and, because the pre-abandonment morphology is cuspate and continuous, no spit forms after abandonment (Fig. 7b) (Nienhuis et al., 2013). For high fluvial sediment supply during growth ($Q_r > 50$ kg s$^{-1}$, $R > 1$), the delta develops a pointy shape and a spit forms after abandonment (Fig. 7b).

Therefore, we estimate that the early cuspate morphology (around 3000 years BP, Canicio and Ibáñez, 1999; Cearreta et al., 2016) was formed with a fluvial sediment supply of at most 35 kg s$^{-1}$. The latter, more rapidly growing Riet Vell lobe that was reworked into a spit, was formed with a fluvial sediment supply of likely more than 50 kg s$^{-1}$. Extending the progradation trajectory of the Riet Vell lobe (Fig. 7c) and keeping in mind that the modern bathymetry suggests a maximum Riet Vell lobe extent of ~20 km

(Canicio and Ibáñez, 1999), we estimate a Riet Vell fluvial sediment supply of ~70 kg s$^{-1}$. Note that these model outcomes are sensitive to model parameters such as the effective shoreface depth, the littoral CERC formula constant and the wave height (Ashton and Giosan, 2011), which were calibrated as described in section 3.1.

### 4.2    Timescales of change on the delta plain

The delta evolution model not only allows us to estimate the morphology of wave-influenced deltas, but also allows us to assess

the timescales Ebro Delta morphologic change. To investigate the timescales, we have simulated the growth and reworking of all three lobes. In 42 different simulations we use the estimated fluvial sediment supply of 70 kg s$^{-1}$ and we vary the growth times of





the different lobes. For example, in one simulation we grew the Riet Vell lobe for 800 years, then the Sol de Riu lobe for 400 years, and finally the Mitjorn lobe for 300 years. In another simulation, we used growth times of respectively 500, 500, and 500 years.

To assess which one of all the 42 simulations best represents the actual history of the Ebro Delta, we measured the radial lengths
of the modeled lobes through time. Then, we measured the radial lengths of the lobes on the modern Ebro Delta from the avulsion apex. Both the paleo channels of the Riet Vell and the Sol de Riu lobe currently extend approximately 10 km from the avulsion apex. The modern active lobe, the Mitjorn, extends about 15 km from the avulsion apex (Fig. 2). The best matched model simulation is the one where the three lobes reach the currently observed lengths of the modern Ebro Delta at the same time. This "reverse engineering" approach yields an estimate of how long each lobe was active and therefore also of the start of Ebro Delta's
rapid growth. These estimates are made independently of published field studies.

For example, in one simulation the Riet Vell lobe grows for 800 years, the Sol de Riu for 400 years, and the Mitjorn for 300 years (dashed lines in Fig. 8). We find that for these growth times the radial extent of the Riet Vell and the Sol de Riu are never 10 km when the Mitjorn is 15 km (the current observed channels lengths) because both the Riet Vell and the Sol de Riu have eroded too
much since their avulsion.

The best matched model scenario of the consecutive growth of the three delta lobes has growth times of 1200, 600 and 300 years, respectively (solid lines in Fig. 8), before it reaches the modern observed lengths of 10 km for the Riet Vell and Sol de Riu and 15 km for the Mitjorn. We estimate therefore, based on this best matched model scenario, that the period of rapid growth of the Ebro
delta lasted 1200+600+300 = 2100 years, placing the time at which rapid Ebro delta growth started approximately 2100 years BP (Fig. 8). These growth times would suggest that the second avulsion occurred 300 years BP, and the first avulsion occurred 900 years BP. Note however that these avulsion times estimates are sensitive to the fluvial sediment supply to the delta (here kept at 70 kg s$^{-1}$) and its variability through time. We keep the fluvial sediment supply constant during the simulations to limit the number of model variables and keep this strictly a scenario-based approach.

The best matched model estimates for the start of rapid delta growth, made purely based on physical constraints set by alongshore sediment transport and fluvial sediment supply, roughly coincide with observations suggesting increased flood plain deposition in the drainage basin (2000-1800 years BP Thorndycraft and Benito, 2006). The model-estimated avulsion times also compare closely with scant historical evidence (Canicio and Ibáñez, 1999; Somoza and Rodriguez-Santalla, 2014), at least for the avulsion of the
Sol de Riu at ~300 years BP. We also find that the maximum extent of the modeled Riet Vell Lobe (~20 km, Fig. 8) approximates earlier indications of its extent made from Ebro Delta bathymetry (Canicio and Ibáñez, 1999). The qualitative agreement between the model scenario and the growth, reworking, and spit formation observed on the Ebro delta, suggests the possibility that the gross morphology of the delta plain can develop without significant sea level or fluvial sediment supply fluctuations.

Importantly, model simulations show the development of spits during both lobe growth and lobe abandonment. However, these spits grow at different orientations (Fig. 8). Ashton et al. (2016) suggest that spit orientation is strongly affected by the updrift shoreline change rate. We speculate that, based upon their more river parallel orientations, the lagoons in the southern region of the modern Ebro delta plain (e.g. the Encanyissada, Clot, and Tancada lagoons, Fig. 2) formed as they were enclosed by spits created while the delta was growing. On the other hand, the active southern La Banya spit has a different orientation because it
was formed as the updrift shoreline retreated during reworking of the Riet Vell lobe.



### 4.3 Wave climate change as a potential cause of delta growth

Investigating the effect of changes in sediment supply on the Ebro delta, we assumed the wave climate was constant. However, previous studies (Goy et al., 2003; Sabatier et al., 2012) focusing on the western Mediterranean over the last millennia suggest evidence exists of changes in wave climate as well. Goy et al. (2003), studying the cuspate coast of the Gulf of Almeria in southern

Spain, correlated beach ridge progradation to periods of negative North Atlantic Oscillation (NAO) because of stronger winds from the southwest that would increase littoral drift to the coast.

To investigate the potential effect of a change in the NAO index on the fluvial dominance ratio $R$, we correlated the monthly NAO index with the Hipocas record, the longest wave climate hind-cast record available spanning 44 years (Table 1). We use a littoral

$K_1$ constant of 0.024 compared to the 0.035 used in the delta evolution model (eq. 1) to correct the alongshore sediment transport predictions of the Hipocas record compared to the Cap Tortosa record (Table 1).

Over this 44 year timespan, there were higher waves from the south during periods of negative NAO (Fig. 9a). For more positive NAO values, average wave height is lower, particularly from the south. Calculating the monthly $Q_{s,max}$, and comparing it to the

NAO index, we find a weak trend from 100 kg s$^{-1}$ (3.2 MT yr$^{-1}$) for strongly negative NAO (-4), to 40 kg s$^{-1}$ (1.2 MT yr$^{-1}$) for periods of strongly positive NAO (+4) (Fig. 9b). We use this obtained trend to assess late Holocene changes in $Q_{s,max}$ based on a NAO index proxy record from the last 2000 years.

Climate reconstructions suggest that the NAO index since the mid Holocene can be divided into three distinct periods. Prior to

2000 years BP the NOA index was mostly negative, afterwards up to about 600 years BP it changed to become mostly positive. Over the past 600 years, the NAO index has been fluctuating with short but strongly negative periods (Chen and Van den Dool, 2003; Olsen et al., 2012).

To obtain approximations of $Q_{s,max}$ for each of the three periods, we determined representative distributions of NAO indices from

NAO paleoclimate records (Fig. 9b). We find that extreme NAO indices are rare and that the distributions of NOA indices, even though distinct, also overlap considerably. Therefore, although $Q_{s,max}$ can vary with changes in the NAO, particularly on a year-to-year basis, geologic constructions of NAO do not suggest significant sustained differences across the previous two millennia (Fig. 9c). This suggests that changes in the fluvial sediment load have likely been a more important driver to the morphodynamic change of the Ebro delta than wave climate changes.

### 30 4.4 Timescales of environmental change in the fluvial catchment

Results from the delta evolution model in concert with previous records indicative of hydrologic change (Thorndycraft and Benito, 2006), place the start of Ebro delta's rapid growth at approximately 2100 years BP. Additionally, CEM model experiments indicate that roughly a sustained doubling in the sediment flux from 35 kg s$^{-1}$ to 70 kg s$^{-1}$ over this period of time could create the observed morphologic changes in growing delta morphology. We have run four different scenarios in the river profile model to estimate the

types and timing of drainage basin changes that could explain this increased fluvial sediment supply to the delta from 35 kg s$^{-1}$ to 70 kg s$^{-1}$ starting 2100 years BP and lasting up to the 20$^{th}$ century. The four scenarios are: (1) an increase in fluvial sediment supply, (2) an increase in fluvial flood discharge, (3) an increase in fluvial flood discharge and fluvial sediment supply, and (4) an increase in fluvial flood discharge and a 500 year lag in an increase in fluvial sediment supply.



In scenario one we change the fluvial sediment supply 450 km upstream from the Ebro delta from about 35 kg s⁻¹ to 70 kg s⁻¹, with the flood discharge and its intermittency remaining constant (Table 2). Such a scenario could arise from land clearing that increased sediment supply without altering the discharge. The model experiment shows that the channel bed slowly aggrades to the new sediment supply and that the change in supply signal takes about 4000 years to significantly affect the Ebro delta (Fig. 10). This

increase is associated with upstream aggradation of about 130 m. While there are numerous field studies that show large alluvial deposits throughout the Ebro drainage basin that date between 6000 years BP up to 2000 years BP (e.g. Benito et al., 2008; Constante-Orrios et al., 2009; Constante et al., 2010; Constante and Peña-Monné, 2009; González-Sampériz and Sopena Vicién, 2002; Gutiérrez-Elorza and Peña-Monné, 1998; Soriano, 1989), the majority of these deposits are on the order of ~10 m thick. The unrealistic magnitude of the predicted aggradation is in part caused by the assumption that floodplain width remains constant,

although the likely formation of a wider floodplain would not greatly affect the sediment supply to the delta. More importantly, the lack of any observed 130 m thick Holocene deposit makes it unlikely that exclusively a fluvial bedload sediment supply increase occurred in the Ebro drainage basin. Even though subsequent erosion of some deposits is likely, a sustained increase in sediment supply should have been accompanied by a sustained high slope and preserved alluviation (Fig. 10b).

In contrast to an increase in fluvial sediment supply, any change in hydrology (flood magnitude and/or flood duration) affects sediment supply to the delta instantaneously. A doubling in the flood magnitude results in a doubling of the fluvial sediment flux delivered to the delta, but would simultaneously cause the channel to start incising upstream (Fig. 10a). Over time, this discharge-driven incision gradually lowers the fluvial sediment flux at the river mouth, returning to the previous value after approximately 8,000 years (Fig. 10c). A concave-down river profile would be diagnostic of an ongoing upstream adjustment to a large increase

in discharge over the past several thousand years. However, as a concave-down river profile is not observed (Fig. 5b), we find it unlikely that an increase in flood discharge and/or duration is the sole cause of increased Ebro delta growth.

In a third scenario, we investigated a simultaneous doubling of upstream sediment supply and discharge. A combined change in sediment supply and discharge instantly doubles the sediment supply at the delta (Fig. 10a). Over time, incision due to discharge

increases is compensated by the aggradation caused by increased fluvial sediment supply (Fig. 10d).

Lastly, the fourth scenario we tested is also a doubling of the upstream sediment supply and discharge, but now we included a 500 yr lag on the sediment flux. Such a scenario could be result of deforestation, where an instantaneous hydrologic signal is followed by a delayed secondary channel slope signal reaching the main stem of the Ebro River. We find that this fourth scenario has a

double peaked effect in deltaic sediment supply. The first (discharge-driven) peak is instantaneous, and the second (sediment-supply-driven) peak is delayed by ~4000 years (Fig. 10a). Combined, such a delay has a small but measurable (~5 m) effect on the fluvial longitudinal profile (Fig. 10e).

Because floodplain aggradation is dependent on the elevation of the channel with respect to the surrounding floodplain (Heller and

Paola, 1996; Schumm and Lichty, 1963), each of the tested scenarios would leave a distinct record in the floodplain deposits. Our forth scenario of increased floods leading to channel incision and a delayed increase sediment flux leading to channel aggradation has not only a double peaked response on the delta, but is also expected to have a double peaked response in floodplain aggradation. Our fluvial profile model suggests that an increase in flood discharge would reflect an initial period of floodplain aggradation, and would decrease gradually as the channel starts to incise (Fig. 10i). The second period of floodplain aggradation would be related

to the aggradation resulting from the increase in fluvial sediment supply. Radiocarbon dating of floodplain aggradation across the





entire Iberian Peninsula similarly shows two periods of increased aggradation in the last 2000 years, one between 2000 and 1830 years BP, and one between 910-500 years BP (Benito et al., 2008).

In general, the river profile model experiments suggest an increase in either sediment or discharge alone are not responsible for the rapid growth of the Ebro Delta. Instead, a combination of increased flood discharge and increased fluvial sediment supply generates a response that best agrees with our understanding and previous findings of changes on the Ebro delta plain. The observed channel bed slope appears to be in a long-term equilibrium, with no evidence of thick Holocene floodplain deposits. These model results here show that changing flooding and sediment discharge by the same amount mostly cancel each other out, resulting in a sustained signal that can be felt instantaneously at the river delta. Both climate change and human impacts on landscapes such as deforestation can increase both the fluvial flood discharge and the fluvial upstream sediment flux (Cosandey et al., 2005; Ferrier et al., 2013), which makes it difficult to use our model results to quantify the individual response of either climate or land-use changes. However, the application of this fluvial profile model does highlight that care should be taken when assuming that any change in the basin can result in an instantaneous and sustained change in sediment delivery to the delta.

## 5    Discussion and conclusions

In this study we used two reduced-complexity models to temporally and physically constrain the late Holocene evolution of the Ebro delta. Where possible, we assumed the simplest possible scenario of environmental change, focusing on the first-order effects on the Ebro River and its delta. Given that we do not find evidence of significant long-term changes in the wave climate, model experiments show that an increase in the coarse fluvial sediment supply to the delta approximately 2100 years BP is the most likely driver of growth of the modern Ebro delta plain, whereby the delta prograded approximately 2-3 times faster than before (Cearreta et al., 2016). Additionally, model experiments with the delta evolution model show that Ebro delta avulsions, where reworking of the abandoned lobes resulted in development of the modern La Banya and El Fangar spits, likely occurred around 900 years BP and 300 years BP, respectively, consistent with previous studies (Canicio and Ibáñez, 1999).

Aside from physically constraining Ebro delta change, our models also highlight the physical mechanisms responsible for the generation of observed morphology. Simulations also point to the formation of spits during delta growth, potentially responsible for delineating the Clot, Encanyissada and Tancada lagoons, with orientations distinct from large recurved La Banya and El Fangar spits that formed from reworking of abandoned lobes. The suggested changes to the Ebro delta leading to the formation of the observed spits is possible under a constant sea level and sediment supply.

Using constraints from the delta evolution model together with a river profile model, we find that a combination of fluvial flood discharge and fluvial sediment supply that started approximately 2100 years BP is the most likely cause of a rapid and sustained period of deltaic growth over the last 2100 years. The rapid growth of the Ebro delta is not solely caused by an increase in fluvial flood discharge because that would greatly increase fluvial incision. Instead, a combined change in discharge and sediment supply can be felt instantaneously at the river delta while persisting for millennia without a significant channel profile change. A combined change in discharge and sediment supply can also, depending on their respective timing, generate an observable floodplain record (Fig. 10i).





In this study we have highlighted a few factors that particularly influence the sensitivity of the results presented here. Fluvial sediment supply, wave climate characteristics, and the littoral sediment transport constant all have a first-order effect on gross delta shape. Shoreface characteristics such as the depth of closure and the basin depth determine how the delta responds to sediment flux changes. Timescales of the river profile model are particularly sensitive to the upstream boundary location: the average

distance between the delta and environmental change in the drainage basin. In all of the simulations presented here, we have chosen average, representative model parameters frequently mentioned in literature, with model results showing the broad first-order agreement with other studies of Ebro Holocene evolution. By quantifying potential effects of historical land-use and climate change on historical delta evolution, simple models might also be able to simulate long term future deltaic change and help guide management decisions (Giosan et al., 2014).

**Acknowledgements**

This study was supported by NSF grant EAR-0952146.

**Competing interests**

The authors declare that they have no conflict of interest.

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



**Tables and Figures**

**Table 1.** Overview of five different sources of wave climate data close to the Ebro delta. See Figure 3 for an overview of locations and the angular distribution of alongshore sediment transport potential. Wave height is the effective, yearly averaged wave height weighted by its ability to move sediment alongshore, i.e. $(\Sigma Hs^{2.4})^{1/2.4}$. The $R^2$ value is the coefficient of determination of the alongshore sediment transport calculated from the wave data versus the measurements of Jiménez and Sánchez-Arcilla (1993).

| Name | Type | Lat °N | Lon °E | Water depth (m) | Wave height (m) | Wave period (s) | Qs,max (kg s$^{-1}$) | $R^2$ | Data period (yr) | Reference |
|---|---|---|---|---|---|---|---|---|---|---|
| Cap Tortosa | buoy | 40.7 | 1.0 | 60 | 0.8 | 4.1 | 47.9 | 0.89 | 1990-2011 | Bolanos et al., 2009 |
| Tarragona | buoy | 41.0 | 1.2 | 24 | 1.0 | 5.5 | 72.4 | 0.86 | 2004-2011 | Puertos del Estado, 2015 |
| MedAtlas | model | 40.0 | 1.0 | 222 | 0.7 | 4.0 | 48.3 | 0.76 | 1992-2002 | Gaillard et al., 2004 |
| Hipocas | model | 40.8 | 1.0 | 68 | 1.1 | 4.9 | 71.1 | 0.87 | 1958-2001 | Sotillo et al., 2005 |
| Wavewatch III® | model | 40.8 | 0.8 | 63 | 0.7 | 4.9 | 31.1 | 0.86 | 1979-2009 | Chawla et al., 2013 |

**Table 2.** Overview of the four river profile model experiments and their final equilibrium slope and bed level change. $Q$ is the fluvial flood discharge, $Q_r$ is the upstream fluvial sediment supply, $i$ is the initial antecedent fluvial environment, and $f$ is the final fluvial environment.

| Description | $Q_i$ (m$^3$s$^{-1}$) | $Q_f$ (m$^3$s$^{-1}$) | $Q_{r,i}$ (kg s$^{-1}$) | $Q_{r,f}$ (kg s$^{-1}$) | Slope (i) (·10$^{-4}$) | Slope (f) (·10$^{-4}$) | Upstream bed level change (m) |
|---|---|---|---|---|---|---|---|
| Sediment x2 | 900 | 900 | 35 | 70 | 2.9 | 5.8 | 130 |
| Discharge x2 | 420 | 900 | 35 | 35 | 5.8 | 2.9 | -130 |
| Discharge and sediment x2 | 420 | 900 | 35 | 70 | 5.8 | 5.8 | 0 |
| Discharge and sediment x2 w/ delay | 420 | 900 | 35 | 70 | 5.8 | 5.8 | 0 |





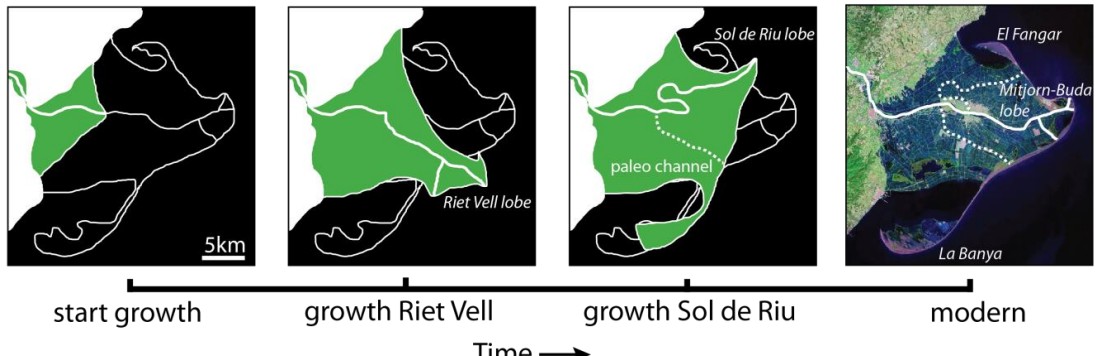

**Figure 1: Reconstructed morphologic development of the Ebro delta, modified from Canicio and Ibáñez (1999).**



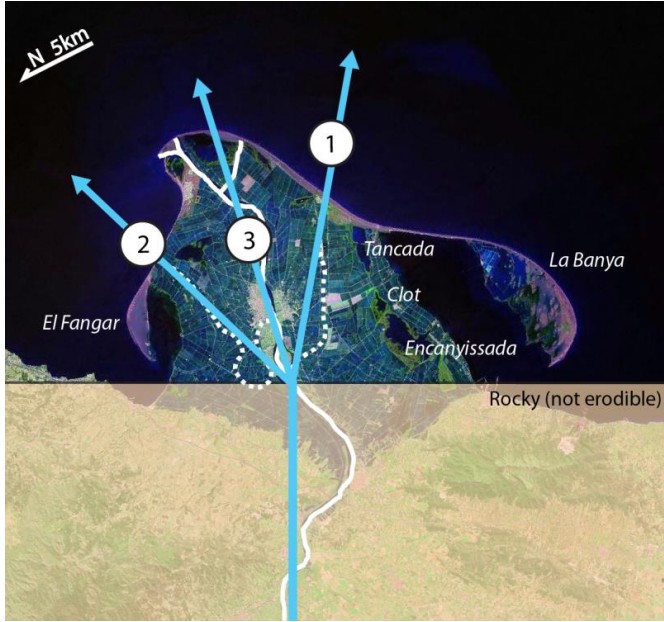

**Figure 2. Schematic of modeling scenario, highlighting the succession and orientation of Ebro delta lobes, shown on top of the modern Ebro delta morphology (NASA Landsat image) and the inferred paleo channels (dotted lines, from Maldonado, 1975). In the model, the straight reference coastline is assumed to be non-erodible. Names refer to the spits and the lagoons on the Ebro delta. Numbers refer to the (1) Riet Vell, (2) Sol de Riu, and the (3) Mitjorn-Buda lobes.**

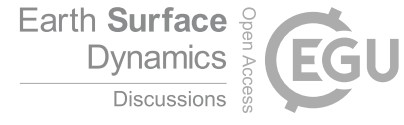

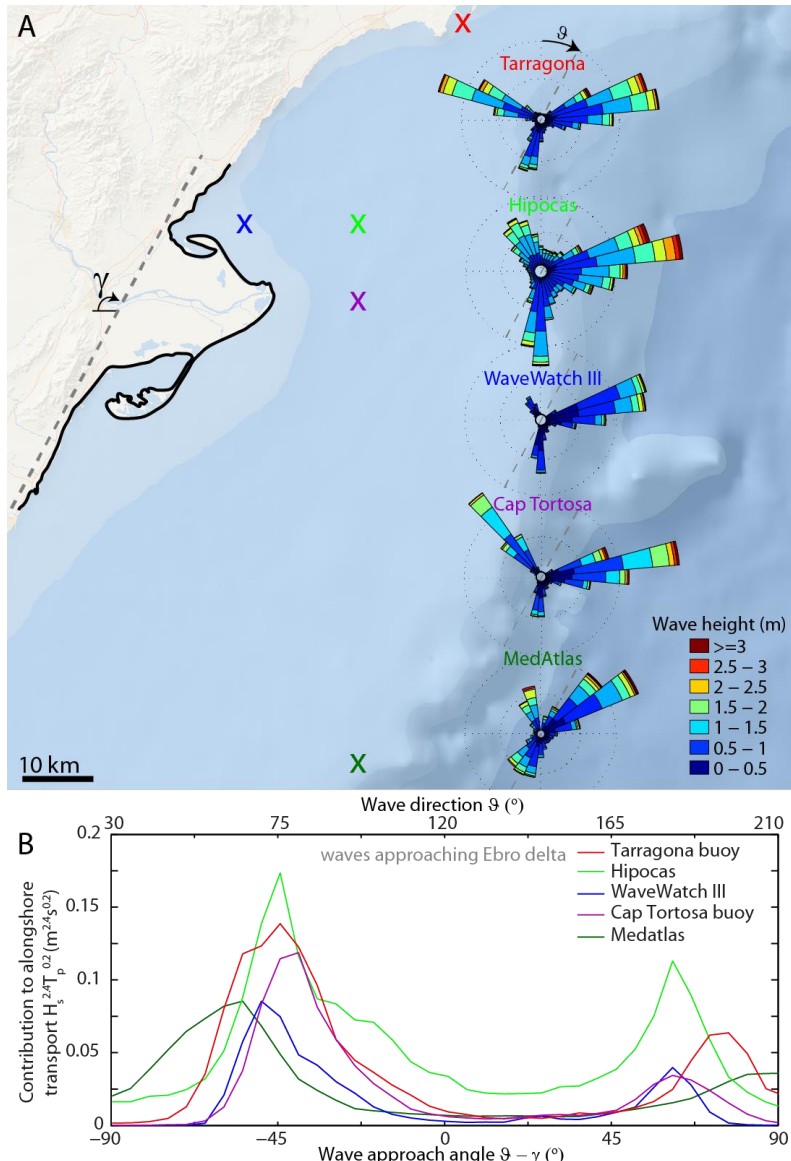

**Figure 3: (A) Comparison of the five different wave roses and their location on a map from NOAA (2015). See table 1 for an overview of the sources. (B) Angular distribution of alongshore sediment transport potential for the five different sources.**



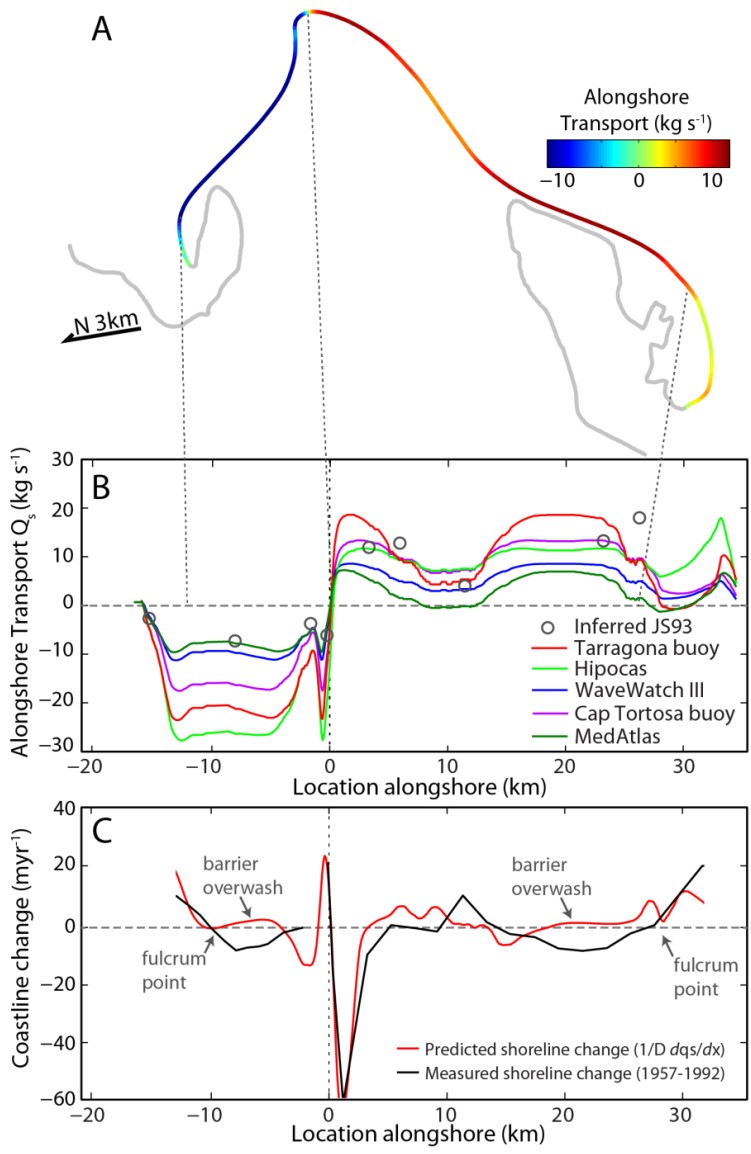

**Figure 4. (A) The Ebro delta coastline, colored by the simulated alongshore sediment transport flux from the Cap Tortosa data. (B) Alongshore sediment transport along the Ebro delta coastline from all five wave climate sources (and assuming no sediment was supplied by the Ebro River). Alongshore transport is positive to the right when looking offshore. Black markers indicate alongshore sediment transport estimates from Jiménez and Sánchez-Arcilla (1993). (C) The Cap Tortosa buoy data recast into shoreline change rates using the one-contour-line approach (eq. 2) compared to the measured shoreline change rates from Jiménez and Sánchez-Arcilla (1993).**





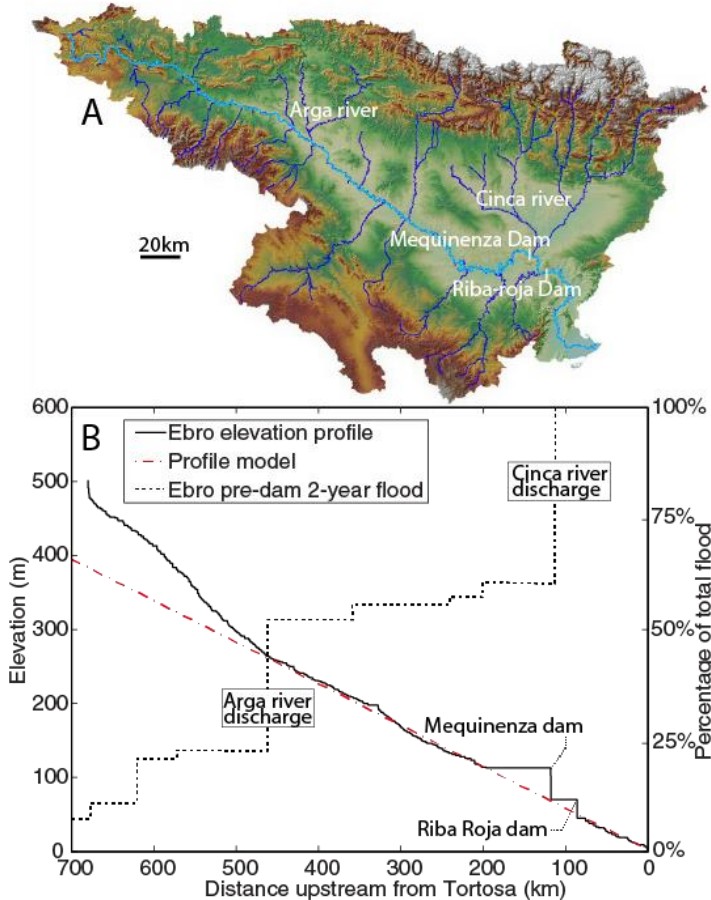

Figure 5. (A) The Ebro river basin showing the main river channel in light blue and larger tributaries in darker blue, colored by elevation. (B) The elevation profile of the Ebro River, with the equilibrium profile model prediction in red dashed line. The black dashed line shows the cumulative fraction of the Ebro pre-dam discharge from Batalla (2004).



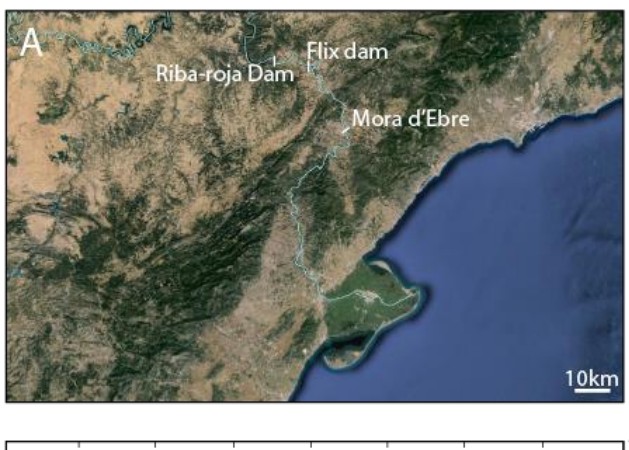

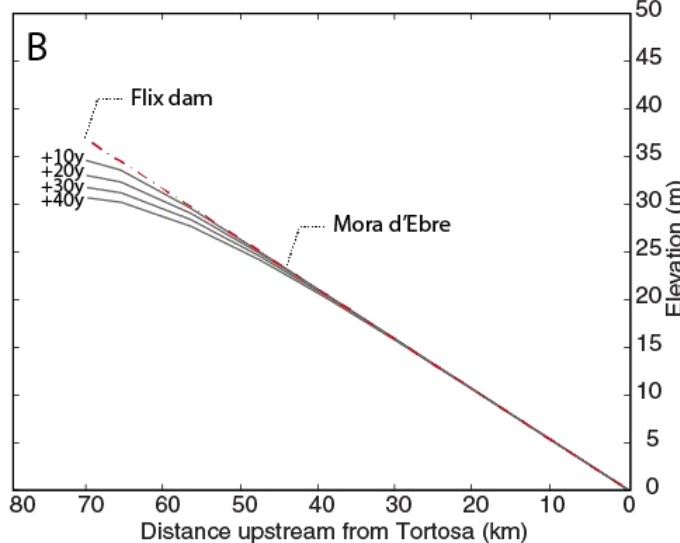

**Figure 6. (A) A close-up of the Ebro drainage basin close to the delta (data from Google Earth, 2015). (B) Modeled response of the Ebro River downstream of the lowermost modern dam, the Flix dam. The bed degradation measurements from Vericat and Batalla (2006), are taken 25 km downstream of the Flix Dam, in Mora d'Ebre.**



Earth **Surface**
**Dynamics**
Discussions



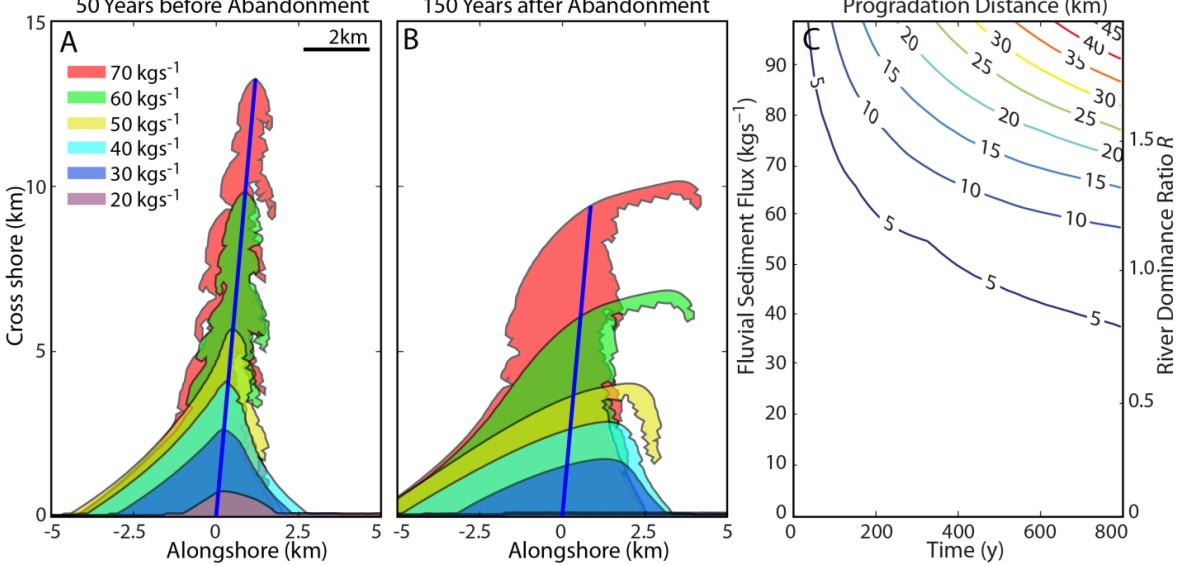

**Figure 7. A modelled delta lobe (A) after 750 years of growth and (B) after 150 years of reworking (950 years of total model time). (C) Contour diagram of the progradation distance versus time as a function of the fluvial sediment flux *Q*ᵣ, or the river dominance ratio *R*.**




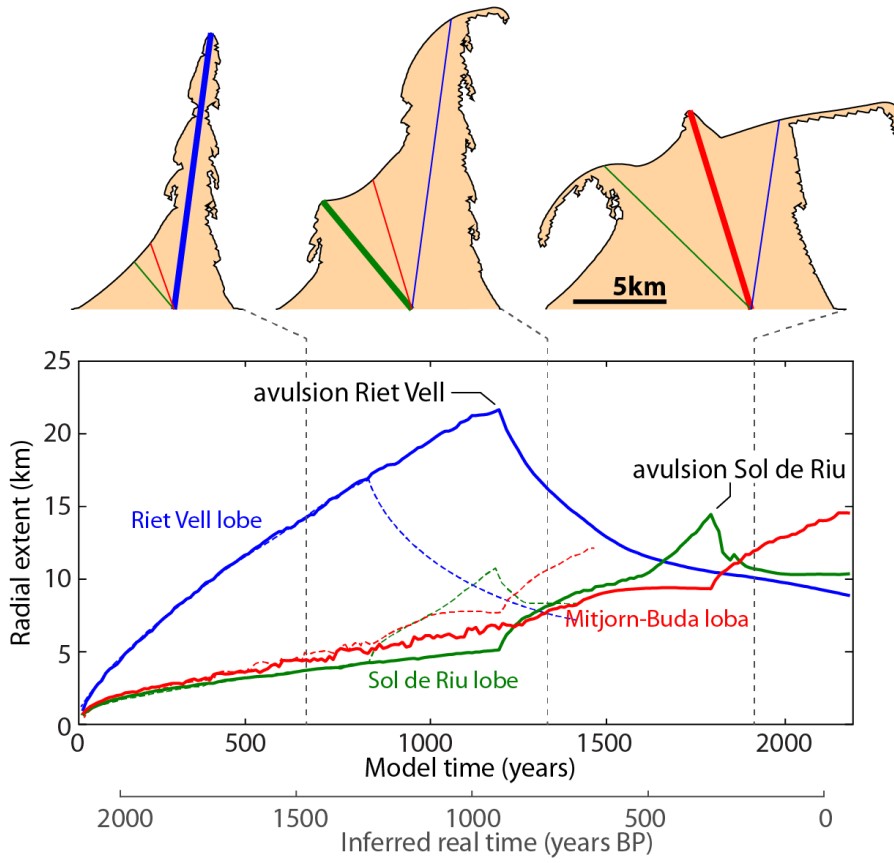

**Figure 8. Simulated radial extent of the three different Ebro delta lobes for a sediment supply of 70 kg s⁻¹ and forced avulsions after 1200 and 1800 model years (solid lines) and after 800 and 1200 model years (dashed lines). Note that the radial extent can increase without the lobe being active because of littoral sediment transported from adjacent lobes. Three inset deltas show the solid line model run after**
5 **700, 1350 and 1900 years. The gray 2ⁿᵈ horizontal axis indicates the real time inferred from the solid line model run and the modern Ebro delta morphology, where at the year 2015, lobes 1 and 2 are approximately 10 km long, and the active lobe is 15 km long, measured from the apex (Fig. 2).**


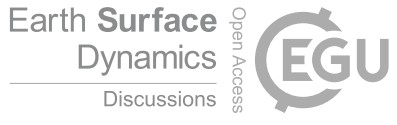

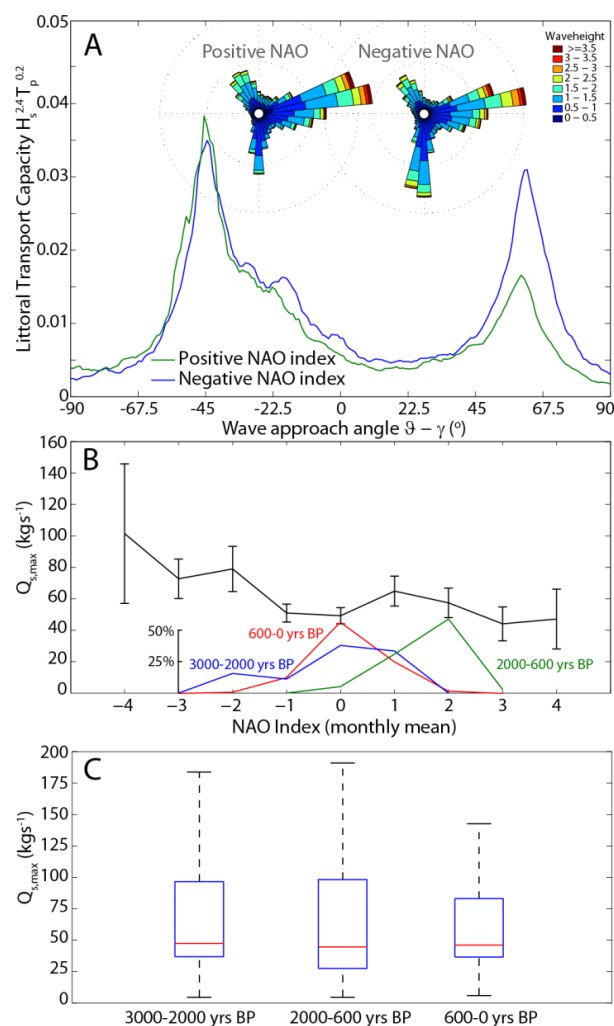

**Figure 9. (A) The angular distribution of alongshore sediment transport potential from Hipocas hind-cast data (Sotillo et al., 2005), separated into periods of negative and positive monthly North Atlantic Oscillation (NAO) index (Chen and Van den Dool, 2003). Insets show wave roses weighted by alongshore sediment transport potential for positive and negative NAO. (B) The effect of the monthly NAO**
5 **on the maximum potential alongshore sediment transport $Q_{s,max}$. Inset shows the NAO index distribution for 3000-2000 years BP, 2000-600 years BP, and 600-0 years BP (from Chen and Van den Dool, 2003; Olsen et al., 2012). (C) Computed distribution of $Q_{smax}$ for different time periods.**





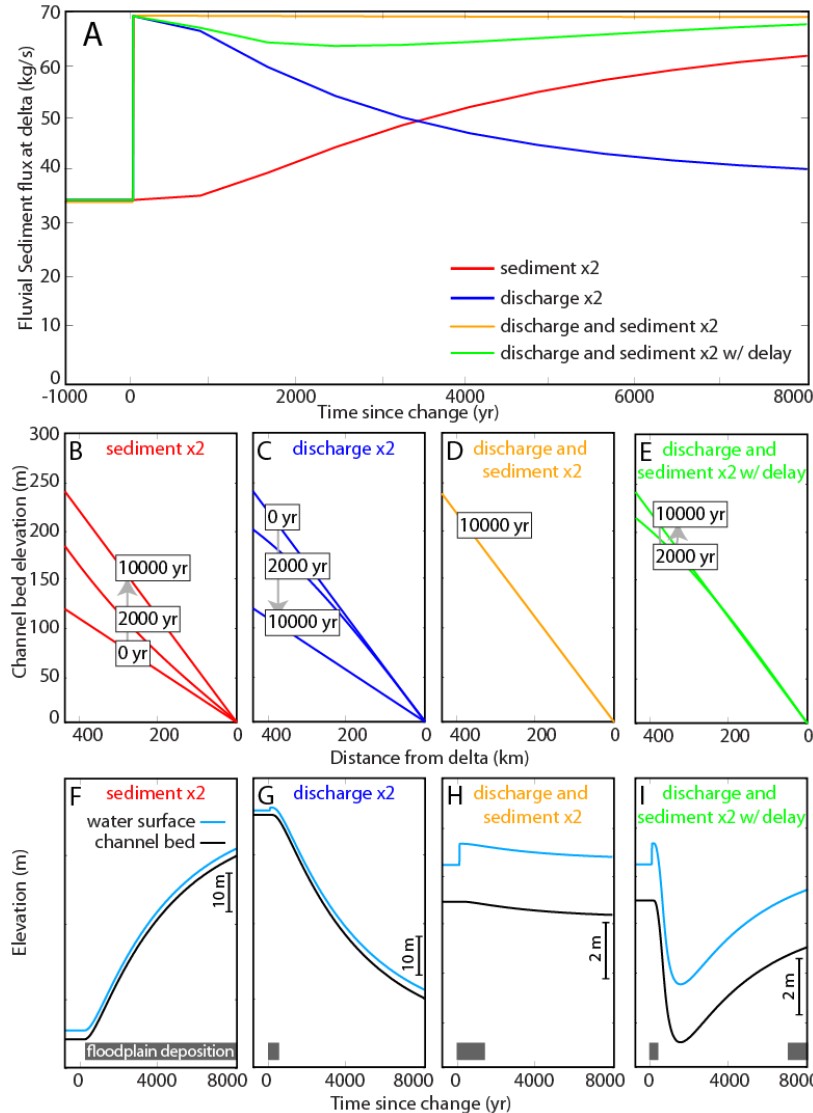

**Figure 10. (A) Fluvial sediment flux at the apex of the delta and (B-E) longitudinal river profile evolution from four experiments of the river profile model with a doubling (x2) in: sediment supply (red), flood discharge (blue), sediment supply and flood discharge (orange), and flood discharge with a lagged sediment supply (green). (F-I) Time evolution of the channel bed and water surface elevation through time, 200 km upstream of the delta at the approximate location of the floodplain records from Benito et al. (2008). Note the different scales between (F-G) and (H-I). Expected occurrence of floodplain deposits shown by the grey bars.**