# Peer review of "Large-scale coastal and fluvial models constrain the late Holocene evolution of the Ebro Delta"

_Earth Surface Dynamics, 2017_

## Referee Comment (RC1) · J. Guillen (Referee) · 27 Mar 2017

General comment: This is a "bold" manuscript exploring how the evaluation of the sediment budget in a coastal system during long-term periods (thousands of years) is suitable for the interpretation of past sedimentary processes, their timing and their morphological evolution (and presumably be applied to future projections). I like this aspect of the work. However, I suspect that the necessary assumptions required to simplify natural processes in the model make the results merely conjectures without firm evidence and that different test proposed by authors are just a sensitivity analysis of considered parameters. In fact, the application of this methodology to the Ebro delta evolution during the late Holocene mainly adjusts model results to previously known

data (or interpretations derived from it). This provides the opportunity to authors to discuss several issues of the Ebro delta recent evolution that are interesting but quite speculative.

Specific comments: Sometimes I'm a little confusing with the use of the term "delta" in the manuscript. The Ebro delta (understood as delta plain, prodelta and associated fluvial and lagoon environments) developed during the Holocene (Díaz et al., 1996), but previous "delta" deposits are recognized before since the Messinian (Farrán and Maldonado, 1990; Urgelés et al., 2011). Sentences as "the delta was already formed -6000 years BP" (p. 4, l 10) or "...the effect of fluvial sediment supply on Ebro delta morphology..." (p. 12, l15) suggest that delta and delta plain are used indistinctly along the text. In fact, a question what comes to my mind is if we can properly reconstruct the Holocene sedimentary history of a deltaic area and their fluvial inputs just using the shoreline variations and almost ignoring the submerged delta (the present-day delta plain area is about 325 km2 and the prodelta area is one order of magnitude larger, about 2300 km2). I realize that the 1-D model of shoreline evolution assume that shoreline variability is proportional to the shoreface translation considering a constant shape of the profile (and the shallowest submerged delta is included in this way). However, previous studies show that the depth of closure varies along the delta and, probably, there were important changes in the littoral profile during progradational and erosional periods of the shoreface. This is corroborated by the distinct morphology and sediment distribution on previously abandoned deltaic lobes areas (Guillén and Palanques, 1997). I am afraid that values obtained from these approximations are very close to the error range of the method because these uncertainties. For instance, it sounds reasonable to expect values of subsidence in the Ebro delta area of a few mm per year. During 2000 years this implies changes of several meters in the level of emerged and submerged delta. Apparently this should be a significant parameter for long-term evolution that probably change the sediment budgets inferred from shoreline data but which is ignored in the manuscript.

Estimation of sedimentary fluvial inputs and fluvial model: Here there is a mesh of data from different sources. To choose a grain size of 0.2 mm for the fluvial profile model seems unrealistic. This sediment grain size characterizes deltaic beaches but the sediment in the river (including in the delta plain) is coarser. Upstream of the deltaic area most of fluvial bed sediment is gravel. The assumption that this sediment (0.2 mm grain size) is mostly transported during floods of 900 m3/s is also inaccurate. Batalla et al (2004) refers this value for bedload of gravel beds upstream of delta plain. The bedload transport in the river at the delta plain (which determines the sediment supplied to the coastal zone) begins with water discharges of about 400 m3/s and progressively increases with water discharge (flow velocity). There is an inflection point in this relation with water discharges around of 800-900 m3/s. This means that the potential bedload transport is "most effective" with that water discharges, but total bedload transport depends of the duration of flow conditions. Finally, the estimated sediment supply of 70 Kg/s-1 during Riet Vell formation and used in model simulations, which is equivalent to the pre-dam bedload flux (71 kg s-1) by Syvitski and Saito (2007), should be considered as a feasible number that could give an order of magnitude of sedimentary inputs but whose variation would significantly change the results of the model.

I found the analysis of section 4.3 about wave climate change during the Holocene really weak. The evaluation of storminess during the Holocene is a complex issue and the approximation carried out in this section is too simplistic to prove any trend.

---

## Referee Comment (RC2) · E. viparelli (Referee) · 27 Mar 2017

This manuscript describes an interesting application of two reduced complexity models to quantitatively characterize the long term impact of changes in flow rate and sediment loads on the progradation of the Ebro delta over the last ∼2000 years. The application of the two models, a coastline evolution model and a river morphodynamic model, is novel in the sense that the output parameters of the river model are used to update the input conditions of the coastline evolution model. Although the models were not fully coupled because the input parameters of the coastal evolution model do not seem to change in time during a simulation, the results of this exercise are useful to determine what could have caused an the increased delta progradation rates that occurred about

∼2000 years ago. I consider the level of model simplification appropriate for the spatial and temporal scales of interest. I like the choice of not modeling autogenic river avulsions and backwater effects and to impose the orientation of the channels based on field observation. The model is well written and I have some general comments on the manuscript and I list them below.

Comment 1

The detailed description of recent changes in flow regime and sediment supply to the delta (section 2.4) is relevant to characterize the present Ebro delta, however this information does not seem to be used in the model application and in the discussion sections of the manuscript. Is the Ebro delta suffering of land losses or shoreline retreat? How are these changes (if they have been documented) related to the dam construction based on the four model scenarios considered in the manuscript?

Comment 2

It is not very clear how the effects of changes in flow regime and sediment supply to the Ebro delta were studied. One of the output parameters of the fluvial model can be the mean annual sediment load (I do not remember if the original model has it as output parameter or if the code needs to be slightly modified to print it). Are the authors imposing a variable sediment supply or its equilibrium value, i.e. the value at the end of the numerical simulation when the system reaches a new equilibrium state? I understand that equilibrium values of sediment supply were used in the simulations. I am not asking to do more simulations, but it can be nice to fully couple the two models in the near future and see how the coastline evolution changes in case of sediment supply that changes in time.

Comment 3

The description of the fluvial model can be improved and refined. I would clarify that since the authors are using a channel model, they consider the bed material only and

do not model washload. In line 37-38 the description of equilibrium is not very clear and should probably be improved by saying that in the absence of subsidence/ uplift and sea level rise, if the flow regime and the sediment supply are constant in time rivers tend to reach a mobile bed equilibrium in which the channel bed elevation does not change in time. If streamwise changes in flow discharge and sediment load are not modeled, at equilibrium the bed slope does not change in space and time and the bed material transport capacity is equal to the mean annual supply of bed material everywhere in the modeled reach (Parker, 2004 and 2008).

On page 7, lines 4-15, the normal flow assumption appears and it is not linked to the rest of the text and this part needs some re-writing. I would reference to De Vries (1965) and/or Parker (2004 – chapter 13) to say that when the time scales of changes in channel bed elevation are long compared to the time scales of the changes in flow characteristics, the flow can be approximated as steady, i.e. the time derivatives of the Saint Venant equations are dropped. This is the quasi-steady approximation, which is at the base of the vast majority of the morphodynamic models. When it is further assumed that the flow is locally uniform, the quasi-steady approximation becomes a quasi-normal approximation and the flow characteristics are computed with the formulation that is implemented in the fluvial model used in this study. Thus, on line 9 the normal flow assumption breaks down when the flow is sufficiently non-uniform, i.e. the spatial changes of the flow have to be considered (not non-steady because steady refers to time and when this is the case you cannot drop the time dependence in the flow equations, as happens for e.g. tidal morphodynamics). There is a huge number of river and delta morphodynamic models that use the quasi-normal approximation for the flow (see e.g. Parker et al., 2008 and Paola et al. 2011 for references) and they have been used to approach the study of a large variety of problems. The choice of the quasi-steady or of the quasi-normal approximation depends on the problem of interest, on the available field data and on how the downstream boundary has to be modeled. I honestly do not think that the use of a quasi-normal approximation is a problem for this particular study.

Page 10 line 25, the authors are using a bedload transport relation for 0.2 mm sand. This requires some justification. Why not to use an Engelund and Hansen formulation (Parker, 2004 bulk load relation chapter) for total (bedload plus suspended) bed material load? The model should allow for it. Further, the change in reference Shields number in equation (3) from 0.047 to 0.0495 suggests that the authors are using the Mayer Peter and Muller bedload relation corrected by Wong (Parker, 2004), but they are not changing the coefficient of the load relation. This is perfectly fine with me, since the authors are obtaining reasonable results, but they should mention it in the text.

Comment 4

It is hard to understand how the intermittency factor was estimated.

Comment 5

Figure 6, does the figure become clearer if the temporal changes in bed elevation (eta – eta_initial) are plotted? Do the authors have one or two field data to add to the figure to show that the model is able to reasonably reproduce the field case?

Comment 6

This is a very personal request, can the authors express the sediment fluxes in million tons per year? It is very hard for me to understand how much sediment is delivered to the cost when the fluxes are given in kilograms per second.

Comment 7

Is there any evidence for a change in flow regime and sediment supply to the fluvial reach and to the delta between 6000 and 3000 years ago? It would be nice to have this information to justify the results of the modeling exercise.

Comment 8

A table with the values and the justification of the model parameters will greatly help.

References

De Vries, M. (1965), Considerations About Non-steady Bed-Load-Transport in Open Channels, Publ. 36, Delft Hydraul. Lab., Delft, Netherlands.

Paola, C., R. R. Twilley, D. A. Edmonds, W. Kim, D. Mohring, G. Parker, E. Viparelli, and V. R. Voller (2011), Natural processes in delta restoration, Annu. Rev. Mar. Sci., 3, 67–91.

Parker, G. (2008), Transport of gravel and sediment mixtures, in: Sedimentation Engineering processes: Measurements, modeling and practice, 3, edited by: Garcia, M. H., ASCE, Reston, VA, 165– 251.

Parker, G. (2004), 1D sediment transport morphodynamics with applications to rivers and turbidity currents. Copyrighted e-book. [Available at http://hydrolab.illino

---

## Referee Comment (RC3) · E. Anthony (Referee) · 3 Apr 2017

E. Anthony (Referee)

anthony@cerege.fr

General comment: This is a fine effort that attempts to combine shoreline processes and fluvial water and sediment discharge to account for the evolution of the Ebro River delta based on reduced complexity models. This combination is a novel approach that needs to be encouraged but it is based on many simplified assumptions that can be called into question. The authors have been quite exhaustive in integrating into their model as many parameters and aspects as possible, but one ends up with the impression that the output has been geared to fit input parameters that are not always well determined. This can be expected given the complexity of delta morphogenesis, interactions between fluvial sediment supply and wave climate, and uncertainties regarding long-term large-scale environmental changes involved in such morphogenesis. These weaknesses should not, however, detract from the utility of the combined simple modeling approach proposed by the authors in this paper.

Specific comments:

1. The evidence on the inception and growth of the Ebro delta is altogether rather scanty to be used as a justification for the stages in delta growth replicated by the combined model, especially for the earlier stages of evolution. The use of the presence of beach ridges as a criterion for affirming that the delta was already extant 6000 years ago seems, in this regard, rather dubious as these forms could simply reflect shoreline reworking by waves.

2. The sediment input and grain-size parameters also need to be reconsidered. The construction phases of the delta are based on the supply of sand-sized sediment to the shore. What justifies the choice of a grain size of 0.2 mm in the river channel, given the much larger size range and the dominance of coarser bedload in the channel?

3. The assumption that the wave climate and storminess in this part of the Mediterranean did not change significantly in the course of the evolution of the Ebro is doubtful. More cautious wording should be used regarding this aspect.

4. The changes in delta plan-shape associated with the successive lobes are based on the fluvial dominance ratio but the input data justifying this ratio are rather poorly constrained, and the authors do not seem to consider morphodynamic feedback between lobe plan shape, wave approach direction and alongshore sediment fluxes, except for the current spits.

5. How do recent post-dam changes in water and sediment discharge fit in with the evolution of the modern delta and with the evolution of the two spits flanking the present channel mouth?

---

## Referee Comment (RC4) · J. A. Jiménez (Referee) · 7 Apr 2017

General comment

This manuscript is a very interesting attempt to reconstruct (explore) the long-term evolution of the Ebro river-delta system by using (relative) simple models. The adopted approach based on using wave and river sediment supply scenarios permits to analyse the potential influence of each factor on delta development and, thus, to reconstruct dominant conditions controlling the Ebro delta development. This gives a great flexibility to the analysis since it permits to practically test any combination of forcings controlling deltaic formation and reduction processes. Although this is a great advantage, it also opens the question on how confident authors are on used (selected) conditions.

In addition to this, the proper selection of models' parameters will control obtained results (delta configuration). This may cause that different combinations of both factors (forcing conditions and parameters' selection) will produce a given response.

Specific comments:

[1] Authors use many times the term "delta" and in other places "delta plain". It will be great to clearly specify which is the target (that apparently it is the deltaic plain).

[2] When describing the suitability of the used models, authors mention that they were validated by comparing predictions of observed changes observed during the last century [page 3, lines 21-23]. However, it is not clear how a model "validated" for a period of few decades (for coastal changes) can be used to predict changes in a time frame of millennia.

[3] In different parts of the paper, authors mention the potential effects of deforestation on river sediment fluxes. However, it is not clear/justified in the text which is the magnitude of the deforestation or land-use changes in the river basin required to produce such increase in sediment load. Moreover, it is not justified if population and land use at the required time (1000 years BP) was enough to produce such deforestation.

[4] Authors make reference to a threshold of 860 m3/s to produce bedload transport in the river [page 5, lines 10-11]. However, previous estimations done in the area for 0.2 mm sediment give a threshold value of 400 m3/s. Is this affecting scenario development?

[5] When presenting the delta evolution model, authors mention that they propagate waves to breaking but the presented alongshore sediment transport formula is given as a function of deep water waves.

[6] The calibrated K value of Jiménez & S-Arcilla (1993) used by authors was obtained by comparing computed sediment transport rates with inferred ones from shoreline changes. To do this, several hypotheses were done, being the depth of closure one of

them (to be about 7 m) to convert shoreline to volume changes. Formally, this K value should be strictly valid to be used under the same conditions. Thus, if it is used with a different depth of closure (e.g. 10 m), same wave action should induce a smaller shoreline change.

[7] It is questionable to use a depth of closure of 10 m for very long-term runs (centennial or millennia time scales). This concept was designed to be used at yearly scales and, when time scale increases, it has been observed that it usually increases (e.g. Hinton & Nicholls, 1998). In the study area, non-published data show that beach profiles along the northern part of the Ebro delta taken 20 years later than the work of Jiménez and Sánchez-Arcilla (1993) presented significant bottom changes at locations deeper than 10 m. Moreover, if authors used the inner shelf bathymetry to identify the extension of ancient lobes (following Canicio and Ibañez, 1999), changes are observed down to 20 m water depth. If depth of closure is increased, deltaic plain growth rates will be smaller for given sediment supply and wave scenarios.

[8] How well wave conditions are correlated with NAO? Authors only mention how transport rates change with NAO positive and negative phases but not how significant (in statistical sense) variations are. This is important to support the hypothesis of no significant change in wave conditions during the period of simulation.

---

## Author Response (AR1)

**(Author responses in italics)**

**J. Guillen (Referee)**

General comment: This is a "bold" manuscript exploring how the evaluation of the sediment budget in a coastal system during long-term periods (thousands of years) is suitable for the interpretation of past sedimentary processes, their timing and their morphological evolution (and presumably be applied to future projections). I like this aspect of the work. However, I suspect that the necessary assumptions required to simplify natural processes in the model make the results merely conjectures without firm evidence and that different test proposed by authors are just a sensitivity analysis of considered parameters. In fact, the application of this methodology to the Ebro delta evolution during the late Holocene mainly adjusts model results to previously known data (or interpretations derived from it). This provides the opportunity to authors to discuss several issues of the Ebro delta recent evolution that are interesting but quite speculative.

We thank Jorge Guillen for his thoughtful review. We are aware that many aspects of the fluvial and coastal model are sensitive to the considered parameters. However, we would like to also stress that in this study we are looking for the simplest possible scenario that would mimic the Ebro's modern observed morphology. Therefore an important conclusion that we draw from this study is that we ARE able to reproduce the general Ebro Delta morphological history even when applying simplified models. Such results do not indicate the absence of any complicating factors; rather, they suggest that one does not necessarily have to appeal to such factors to explain the large scale morphology of the Ebro delta.

**We now highlight this important point in the abstract, introduction and conclusion.**

Specific comments: Sometimes I'm a little confusing with the use of the term "delta" in the manuscript. The Ebro delta (understood as delta plain, prodelta and associated fluvial and lagoon environments) developed during the Holocene (Díaz et al., 1996), but previous "delta" deposits are recognized before since the Messinian (Farrán and Maldonado, 1990; Urgelés et al., 2011). Sentences as "the delta was already formed -6000 years BP" (p. 4, 110) or ". . .the effect of fluvial sediment supply on Ebro delta morphology. . ." (p. 12, 115) suggest that delta and delta plain are used indistinctly along the text. In fact, a question what comes to my mind is if we can

properly reconstruct the Holocene sedimentary history of a deltaic area and their fluvial inputs just using the shoreline variations and almost ignoring the submerged delta (the presentday delta plain area is about 325 km2 and the prodelta area is one order of magnitude larger, about 2300 km2). I realize that the 1-D model of shoreline evolution assume that shoreline variability is proportional to the shoreface translation considering a constant shape of the profile (and the shallowest submerged delta is included in this way). However, previous studies show that the depth of closure varies along the delta and, probably, there were important changes in the littoral profile during progradational and erosional periods of the shoreface. This is corroborated by the distinct morphology and sediment distribution on previously abandoned deltaic lobes areas (Guillén and Palanques, 1997). I am afraid that values obtained from these approximations are very close to the error range of the method because these uncertainties. For instance, it sounds reasonable to expect values of subsidence in the Ebro delta area of a few mm per year. During 2000 years this implies changes of several meters in the level of emerged and submerged delta. Apparently this should be a significant parameter for long-term evolution that probably change the sediment budgets inferred from shoreline data but which is ignored in the manuscript.

First, we agree that in many instances we used the word delta when delta plain would be more appropriate and we did so in the revised version.

We also agree that an important simplification in our model approach is the 1-line shoreline assumption. Unfortunately, even though 2 or n-line shoreline models exist, we argue that the application of more complicated models would lead to an increase in uncertainties because of the lack of proper data constraints. Our modeling approach does account for growth of the delta into a deeper basin and subsequent erosion that is limited by the depth of closure. We state this more clearly in the revised version.

Even though subsidence on the delta plain is significant on millennial timescales, we argue that subsidence mostly affects the finer-grained and organic delta topset. We now include a cautionary note concerning the potential uncertainty arising from long-term subsidence of the delta plain.

Regarding the reconstruction of the Holocene sedimentary volume based on the reviewer's delta plain area (325 km2), the shoreface depth we have used (10 m), and the 70 kgs-1 sediment flux we have estimated from the shoreline model:

$$Delta \ age = \frac{DeltaArea(m^2) \cdot ShorefaceDepth(m)}{SedimentLoad(m^3 yr^{-1})} = \frac{325 \cdot 10^6 m^2 \cdot 10m}{70 kg s^{-1} \frac{1}{1600 kg m^{-3}}} = 2355 yr$$

This number is remarkably similar to our model estimate of 2100 years. This first-order calculation does not include the wave dominated delta area older than 2100 years, which we suggest was formed before an increase in fluvial sediment flux. We now include a short section regarding this simple mass balance in the text.

Estimation of sedimentary fluvial inputs and fluvial model: Here there is a mesh of data from different sources. To choose a grain size of 0.2 mm for the fluvial profile model seems unrealistic. This sediment grain size characterizes deltaic beaches but the sediment in the river (including in the delta plain) is coarser. Upstream of the deltaic area most of fluvial bed sediment is gravel. The assumption that this sediment (0.2 mm grain size) is mostly transported during floods of 900 m3/s is also inaccurate. Batalla et al (2004) refers this value for bedload of gravel beds upstream of delta plain. The bedload transport in the river at the delta plain (which determines the sediment supplied to the coastal zone) begins with water discharges of about 400 m3/s and progressively increases with water discharge (flow velocity). There is an inflection point in this relation with water discharges around of 800-900 m3/s. This means that the potential bedload transport is "most effective" with that water discharges, but total bedload transport depends of the duration of flow conditions. Finally, the estimated sediment supply of 70 Kg/s-1 during Riet Vell formation and used in model simulations, which is equivalent to the pre-dam bedload flux (71 kg s-1) by Syvitski and Saito (2007), should be considered as a feasible number that could give an order of magnitude of sedimentary inputs but whose variation would significantly change the results of the model.

We use 900  $m^3 s^{-1}$  as a formative flood at which most of the bed material load is transported. We then adjust the duration (intermittency) of the flow such that the annual bed material load is consistent with the bed material load under the full flow-exceedance frequency curves of Batalla et al (2004) and Vericat and Batalla (2006). We now include a more thorough description of the fluvial profile model parameters in the methods section.

We recognize that the sand designation is inconsistent with the bed material of the Ebro River channel, and we changed bed material in the fluvial profile model to the most transported bedload grain size of 10 mm (Vericat and Batalla, 2006). Because we simultaneously change the channel bed roughness set to reflect a gravel bedded channel (3x D50 of bed material), the resulting fluvial profile timescales and sediment fluxes are similar to the original calculations for a sand bedded river.

We are aware of the simplification of using one fluvial sediment flux to represent an "average" for the last ~2000 years of the Ebro Delta. Variability in this sediment flux would greatly affect the morphologic development of the Ebro Delta. However, we are not aware of any data constraining the fluvial sediment flux history over the previous millenia. An important conclusion of our study is that we are able to reproduce certain elements of the Ebro morphology with the simplest possible assumption: a steady sediment flux.

I found the analysis of section 4.3 about wave climate change during the Holocene really weak. The evaluation of storminess during the Holocene is a complex issue and the approximation carried out in this section is too simplistic to prove any trend.

We agree that paleo wave climate of the Ebro Delta is a complex issue that needs to be investigated and therefore we provide a start for that in our paper. However, based on the most complete NAO reconstruction (Olsen et al., 2012) and the longest wave climate record available for the Ebro Delta (Sotillo et al., 2005) our conclusion is that there appears to be no clear evidence to expect significant wave climate changes, particularly in comparison to previously inferred fluvial sediment flux changes of the Ebro delta and other Mediterranean deltas (Anthony et al., 2014; Giosan et al., 2012; Maselli and Trincardi, 2013). The assumptions underlying our analysis are clearly laid out, including correlations between measured and paleorecords. The lack of strong evidence from our one analysis does not prove that wave climate changes did not occur. However, it does show that more evidence would need to be presented do demonstrate a secular change in wave climate, as has been suggested by others. Also, our results indicate that one does not necessarily have to (or should) appeal to wave climate changes to explain large scale morphologic features of the Ebro delta.

**E. Viparelli (Referee)**

This manuscript describes an interesting application of two reduced complexity models to quantitatively characterize the long term impact of changes in flow rate and sediment loads on the progradation of the Ebro delta over the last ~2000 years. The application of the two models, a coastline evolution model and a river morphodynamic model, is novel in the sense that the output parameters of the river model are used to update the input conditions of the coastline evolution model. Although the models were not fully coupled because the input parameters of the coastal evolution model do not seem to change in time during a simulation, the results of this exercise are useful to determine what could have caused an the increased delta progradation rates that occurred about ~2000 years ago. I consider the level of model simplification appropriate for the spatial and temporal scales of interest. I like the choice of not modeling autogenic river avulsions and backwater effects and to impose the orientation of the channels based on field observation. The model is well written and I have some general comments on the manuscript and I list them below.

**Comment 1**

The detailed description of recent changes in flow regime and sediment supply to the delta (section 2.4) is relevant to characterize the present Ebro delta, however this information does not seem to be used in the model application and in the discussion sections of the manuscript. Is the Ebro delta suffering of land losses or shoreline retreat? How are these changes (if they have been documented) related to the dam construction based on the four model scenarios considered in the manuscript?

We thank Enrica Viparelli for her thoughtful review of our manuscript. Our main answer to this question is that the question is one of scale and objective. Our main objective is to investigate fluvial sediment flux increases that have been hypothesized to have caused the Ebro land gain over the last 2000 years. We do use the recent changes to the Ebro delta shoreline as a test case for the applicability of the coastline modeling, finding that there appears to be little fluvial sediment reaching the modern shoreline. However, our fluvial model scenarios focuses on delta formation. We are hesitant to use the shoreline model or the fluvial profile model to project future changes over the coming century to the Ebro delta because, for one, this has been done previously (e.g., Sánchez-Arcilla, et al, 2008) using a model approach more directly tuned to the historical changes of the delta shoreline. Second, as we now also state, our modeling approach does not account for sea-level rise; projections for the coming century suggest rates that are unprecedented compared to the previous 2,000 year period we investigate. We now state the objective of our research and suggest some potential methods for investigations of future change in the conclusion section of the manuscript.

**Comment 2**

It is not very clear how the effects of changes in flow regime and sediment supply to the Ebro delta were studied. One of the output parameters of the fluvial model can be the mean annual sediment load (I do not remember if the original model has it as output parameter or if the code needs to be slightly modified to print it). Are the authors imposing a variable sediment supply or its equilibrium value, i.e. the value at the end of the numerical simulation when the system reaches a new equilibrium state? I understand that equilibrium values of sediment supply were used in the simulations. I am not asking to do more simulations, but it can be nice to fully couple the two models in the near future and see how the coastline evolution changes in case of sediment supply that changes in time.

We impose its final equilibrium value at the upstream boundary condition, which we now state clearly in the methods and the results section. We impose a constant fluvial sediment supply to the Ebro delta for the coastline model. We chose not to couple the two models directly because they are exploratory and there is no feedback (i.e., the coastal model does not affect the fluvial model). An example of a coupling of fluvial and coastal models is described in: Ashton et al., Comp. and Geosc. 2013. We agree that modeling wave-affected delta evolution with a gradually time-varying sediment influx remains an interesting problem. Here, we use the fluvial models to investigate how a sudden and sustained increase in fluvial sediment input to the shoreline could have affected the evolution of the Ebro delta.

**Comment 3**

The description of the fluvial model can be improved and refined. I would clarify that since the authors are using a channel model, they consider the bed material only and do not model

washload. In line 37-38 the description of equilibrium is not very clear and should probably be improved by saying that in the absence of subsidence/ uplift and sea level rise, if the flow regime and the sediment supply are constant in time rivers tend to reach a mobile bed equilibrium in which the channel bed elevation does not change in time. If streamwise changes in flow discharge and sediment load are not modeled, at equilibrium the bed slope does not change in space and time and the bed material transport capacity is equal to the mean annual supply of bed material everywhere in the modeled reach (Parker, 2004 and 2008).

We realize that the description of equilibrium was less clear and we adjusted the wording following the reviewer's recommendations.

On page 7, lines 4-15, the normal flow assumption appears and it is not linked to the rest of the text and this part needs some re-writing. I would reference to De Vries (1965) and/or Parker (2004 – chapter 13) to say that when the time scales of changes in channel bed elevation are long compared to the time scales of the changes in flow characteristics, the flow can be approximated as steady, i.e. the time derivatives of the Saint Venant equations are dropped. This is the quasisteady approximation, which is at the base of the vast majority of the morphodynamic models. When it is further assumed that the flow is locally uniform, the quasi-steady approximation becomes a quasi-normal approximation and the flow characteristics are computed with the formulation that is implemented in the fluvial model used in this study. Thus, on line 9 the normal flow assumption breaks down when the flow is sufficiently non-uniform, i.e. the spatial changes of the flow have to be considered (not non-steady because steady refers to time and when this is the case you cannot drop the time dependence in the flow equations, as happens for e.g. tidal morphodynamics). There is a huge number of river and delta morphodynamic models that use the quasi-normal approximation for the flow (see e.g. Parker et al., 2008 and Paola et al. 2011 for references) and they have been used to approach the study of a large variety of problems. The choice of the quasi-steady or of the quasi-normal approximation depends on the problem of interest, on the available field data and on how the downstream boundary has to be modeled. I honestly do not think that the use of a quasi-normal approximation is a problem for this particular study.

We thank the reviewer for her recommendations. We have improved the description of the normal flow assumption.

Page 10 line 25, the authors are using a bedload transport relation for 0.2 mm sand. This requires some justification. Why not to use an Engelund and Hansen formulation (Parker, 2004 bulk load relation chapter) for total (bedload plus suspended) bed material load? The model should allow for it. Further, the change in reference Shields number in equation (3) from 0.047 to 0.0495 suggests that the authors are using the Mayer Peter and Muller bedload relation corrected by Wong (Parker, 2004), but they are not changing the coefficient of the load relation. This is perfectly fine with me, since the authors are obtaining reasonable results, but they should mention it in the text.

In the original formulation of the 0.2 mm sediment bed we agree that the Engelund and Hansen formulation would have been more appropriate. However, see also our response to reviewer #3, we changed the sediment bed grain size to 10 mm to more accurately reflect incision and aggradation of the Ebro River. We use the MPM formulation to predict river profile changes with a 10 mm bed median grain size (Vericat and Batalla, 2006).

**Comment 4**

It is hard to understand how the intermittency factor was estimated.

Data (Vericat and Batalla 2006) show that the 900  $m^3 s^{-1}$  flow transports most of the sand and is exceeded ~15% of the time (pre-dam). However, this does not mean that 15% intermittency is a good indicator of annual sediment transport, stronger flows will carry relatively more sediment. An appropriate intermittency is therefore likely higher than 15%.

We use the sediment (which roughly scales with  $Q^2$ ) -exceedance frequency table from Batalla et al (2004) and Vericat and Batalla (2006) to fit a relationship between exceedance frequency and Q2 (the trend is approximately ~  $x^{-1/2}$ ). We then integrate this relationship between 0 and the exceedance frequency of a 700 m3s-1 flow (the minimum to transport bed material) to obtain a sum with units (m3s-1)2 days/year. This sum divided by the discharge2 we use in the model (9002) is about 100 days/year (~30% intermittency). Because we scale the intermittency to the discharge we choose in the model, the results are not very sensitive to an exact flood discharge (e.g. a 1200 m3s-1 flood discharge in the model would correspond to a  $\sim$ 20% intermittency and results in a similar annual sediment flux).

We agree that this was insufficiently clear in the previous text (also pointed out by reviewer #1) and explain this more thoroughly in the methods section of the revised version.

**Comment 5**

Figure 6, does the figure become clearer if the temporal changes in bed elevation (eta – eta\_initial) are plotted? Do the authors have one or two field data to add to the figure to show that the model is able to reasonably reproduce the field case?

We have added a bed level difference plot to this figure, which is indeed more illustrative. We are unfortunately not aware of any channel bed degradation studies of the Ebro River downstream of the dams, aside from the cited studies of Vericat and Batalla (2006)

**Comment 6**

This is a very personal request, can the authors express the sediment fluxes in million tons per year? It is very hard for me to understand how much sediment is delivered to the cost when the fluxes are given in kilograms per second.

We included the MT/yr conversion in every first instance in every section. We omitted it in some instances of high repetition.

**Comment 7**

Is there any evidence for a change in flow regime and sediment supply to the fluvial reach and to the delta between 6000 and 3000 years ago? It would be nice to have this information to justify the results of the modeling exercise.

Thorndycraft & Benito 2006 QSR discuss changes in fluvial flooding in Spain before 3000 years BP. We now discuss their findings in more detail in the background section of the manuscript.

**Comment 8**

A table with the values and the justification of the model parameters will greatly help.

We now include a table providing river profile model and coastal model parameters (new Table 1).

**E. Anthony (Referee)**

General comment: This is a fine effort that attempts to combine shoreline processes and fluvial water and sediment discharge to account for the evolution of the Ebro River delta based on reduced complexity models. This combination is a novel approach that needs to be encouraged but it is based on many simplified assumptions that can be called into question. The authors have been quite exhaustive in integrating into their model as many parameters and aspects as possible, but one ends up with the impression that the output has been geared to fit input parameters that are not always well determined. This can be expected given the complexity of delta morphogenesis, interactions between fluvial sediment supply and wave climate, and uncertainties regarding long-term large-scale environmental changes involved in such morphogenesis. These weaknesses should not, however, detract from the utility of the combined simple modeling approach proposed by the authors in this paper.

We thank E. Anthony for the thoughtful review. We agree with the essence of these comments, that although we explore many parameters in our study, we cannot exhaustively investigate the full breadth of parameter space, in particular the potential for all fluctuations. We hope that we are clear in our assumptions, and emphasize that we do explore a rather large variation in several model parameters that are difficult to constrain (Figure 7). We now more clearly stress the variability and the objectives of the paper in the abstract, introduction, and discussion sections.

**Specific comments:**

1. The evidence on the inception and growth of the Ebro delta is altogether rather scanty to be used as a justification for the stages in delta growth replicated by the combined model, especially for the earlier stages of evolution. The use of the presence of beach ridges as a criterion for affirming that the delta was already extant 6000 years ago seems, in this regard, rather dubious as these forms could simply reflect shoreline reworking by waves.

We agree that beach ridges are not necessarily evidence of delta existence. However, the preserved beach ridges on the Ebro delta plain older than 2000 years are updrift of the mouth curving toward the mouth, in a region where the coast is otherwise cliffed (i.e. no outside sediment sources). This essentially rules out any other possibility but a delta. We

rephrased this section and provide a better summary of the findings presented by Canicio and Ibanez (1997) and Cearreta et al. (2016)

2. The sediment input and grain-size parameters also need to be reconsidered. The construction phases of the delta are based on the supply of sand-sized sediment to the shore. What justifies the choice of a grain size of 0.2 mm in the river channel, given the much larger size range and the dominance of coarser bedload in the channel?

We realize we did not provide sufficient clarification for the 0.2 mm fluvial sediment size. Reviewers #1 and #2 also requested more clarification. We initially used this grain size because it is approximately the grain size of the littoral zone and therefore most likely makes up for most of the delta between the depth of closure and the surface.

However, in response to all reviewers, we now calculate the river profile changes with a 10 mm grain size, the median bed-load grain size (Vericat and Batalla, 2006). We have adjusted the methods section. Because of the likely higher bed roughness ( $\sim 3x D_{50}$  of the median bed material for no bedforms, see Parker 2004) of the gravel bed, the actual sediment transport magnitude is similar to our original results.

3. The assumption that the wave climate and storminess in this part of the Mediterranean did not change significantly in the course of the evolution of the Ebro is doubtful. More cautious wording should be used regarding this aspect.

We state this claim more carefully, also following similar concerns of reviewer #1. However, we used the best data available to us and found no indication that wave climate change would have a strong effect, as significant as changes in the fluvial sediment flux found for the Ebro delta and other Mediterranean deltas (Anthony et al., 2014; Giosan et al., 2012; Maselli and Trincardi, 2013). This of course does not prove that wave climate changes did not occur; rather, it shows that one does not necessarily have to (or should) appeal to wave climate changes to explain some morphologic features of the Ebro delta. We stress this distinction more clearly in the revised manuscript.

4. The changes in delta plan-shape associated with the successive lobes are based on the fluvial dominance ratio but the input data justifying this ratio are rather poorly constrained, and the

authors do not seem to consider morphodynamic feedback between lobe plan shape, wave approach direction and alongshore sediment fluxes, except for the current spits.

The fluvial dominance ratio as we use it in the study purely serves as a diagnostic tool for our model simulations. The simulations themselves include feedbacks between shoreline shape, wave approach angle, and fluvial sediment fluxes. We rephrased our explanation of the fluvial dominance ratio and its use in our analysis in the background and results sections.

5. How do recent post-dam changes in water and sediment discharge fit in with the evolution of the modern delta and with the evolution of the two spits flanking the present channel mouth?

While it is tempting to run the shoreline evolution model into the future we chose not to do so. See also our response to reviewer #2, Enrica Viparelli. There are more appropriate models and methods (e.g. Jimenez 1997) available to predict future delta shape that can take into account a more constrained set of boundary conditions such as (perhaps importantly) sea level rise. A modern Ebro delta model would also probably use the Ebro's present shape as initial condition rather than our assumed shape (a straight coastline) 2000 years ago. We mention this now in the conclusion. However, our analysis of predicted alongshore sediment transport trends and scenario-based models both suggest river damming effects on the shoreline should be noticeable only close to the modern river mouth itself.

**Jose Jimenez (Referee)**

**General comment**

This manuscript is a very interesting attempt to reconstruct (explore) the long-term evolution of the Ebro river-delta system by using (relative) simple models. The adopted approach based on using wave and river sediment supply scenarios permits to analyse the potential influence of each factor on delta development and, thus, to reconstruct dominant conditions controlling the Ebro delta development. This gives a great flexibility to the analysis since it permits to practically test any combination of forcings controlling deltaic formation and reduction processes. Although this is a great advantage, it also opens the question on how confident authors are on used (selected) conditions. In addition to this, the proper selection of models' parameters will control obtained results (delta configuration). This may cause that different combinations of both factors (forcing conditions and parameters' selection) will produce a given response.

We thank Jose Jimenez for his thoughtful review of our manuscript. The two model outcomes are indeed sensitive to many less well constrained variables including the factor k in the shoreline evolution model and the location of the upstream fluvial model boundary. Where possible, we used site-calibrated constants and test model outcomes to recent Ebro river and delta changes. Rather than proving a particular depositional history of the Ebro, we state that our models have provided us with the "best estimate" outcome. Given that this "best estimate" outcome is similar to outcomes from other Ebro delta studies, we can suggest that perhaps our simple models and boundary conditions are sufficient to explain general aspects of Ebro delta morphology that can advance our interpretation on geological timescales. That does not mean that they should substitute for detailed models on shorter timescales.

Other reviewers have brought up similar concerns and we now more thoroughly discuss the sensitivities of the models to specific model parameters. We also more clearly describe the implication of our model results.

Specific comments and responses are discussed below.

Specific comments:

[1] Authors use many times the term "delta" and in other places "delta plain". It will be great to clearly specify which is the target (that apparently it is the deltaic plain).

*Reviewer #1 shared similar concerns. We now more carefully distinguish between the terms delta plain and delta.*

[2] When describing the suitability of the used models, authors mention that they were validated by comparing predictions of observed changes observed during the last century [page 3, lines 21-23]. However, it is not clear how a model "validated" for a period of few decades (for coastal changes) can be used to predict changes in a time frame of millennia.

We agree that perhaps another terminology than the term "validated" is more appropriate. In lines 21-23 of the original manuscript we did not use this specific term, instead stating: "we compare the model predictions to observed deltaic and fluvial change over the last century". However, beyond word definitions, we sought to develop the best possible parameterization, in particular for the alongshore transport coefficient k, and thus used changes over the last century as a calibration. The physics of our model, however, are not dependent on this specific calibration (although this comparison shows that many changes can be explained through alongshore transport gradients). Temporal scale mismatch is not unique to the model we use here; for example, small-scale laboratory experiments are used to calibrate sediment transport parameters used in Delft3D which are often applied over large space and time scales. We believe that several decades represents a reasonable timeframe for testing our model, annual to interannual changes would certainly be inappropriate in this regard. We also do not intend to present our results as an exact replication of the formation of the Ebro Delta.

[3] In different parts of the paper, authors mention the potential effects of deforestation on river sediment fluxes. However, it is not clear/justified in the text which is the magnitude of the deforestation or land-use changes in the river basin required to produce such increase in sediment load. Moreover, it is not justified if population and land use at the required time (1000 years BP) was enough to produce such deforestation.

The sediment flux response to deforestation and land-use changes is an area of active research even in the case of modern changes. We are unaware of detailed studies of the

relationship between historical land-use changes and sediment loads for the Ebro River. Studies such as ours could potentially inform these relationships.

A global compilation of recently modified deltaic systems indicates that increases in the sediment load due to deforestation and land-use changes could amount to a factor of two or larger (Syvitski and Milliman 2007, p 12-13). Xing et al (2014) estimated a 40% increase in the fluvial suspended sediment flux for the Ebro River. We now include a discussion of their results to justify the potential increase of the Ebro river sediment load in response to deforestation.

[4] Authors make reference to a threshold of 860 m3/s to produce bedload transport in the river [page 5, lines 10-11]. However, previous estimations done in the area for 0.2 mm sediment give a threshold value of 400 m3/s. Is this affecting scenario development?

See also our response to other reviewers; as we have changed the median grain size of the bed-material load to 10 mm to more accurately reflect the gravel bed nature of the Ebro River. We now more thoroughly describe how the exact flood discharge of the Ebro is not critically important, as we adjust the flood intermittency to capture the full hydrograph from Batalla et al (2006). We have adjusted the methods section to more clearly describe this procedure.

[5] When presenting the delta evolution model, authors mention that they propagate waves to breaking but the presented alongshore sediment transport formula is given as a function of deep water waves.

We thank Jose Jimenez for pointing out this potential source of confusion. Deep-water wave relationships are used for visualization and analysis of wave climate data (Ashton and Murray 2006 pt. 2). The description of wave breaking is applicable to the numerical 1-line model (Ashton and Murray pt. 1).

[6] The calibrated K value of Jiménez & S-Arcilla (1993) used by authors was obtained by comparing computed sediment transport rates with inferred ones from shoreline changes. To do this, several hypotheses were done, being the depth of closure one of them (to be about 7 m) to convert shoreline to volume changes. Formally, this K value should be strictly valid to be used

under the same conditions. Thus, if it is used with a different depth of closure (e.g. 10 m), same wave action should induce a smaller shoreline change.

We agree with the reviewer's assessment and now add that for the direct comparison between the different wave climates and the shoreline analysis of Jimenez et al. (Fig. 4) we use 7 m as the shoreface depth. The actual model simulations run over much longer timescales, and therefore we use a deeper shoreface depth (10 m). We now include this distinction in our methods section. We refer to our response to detailed comment #7 below for further explanation.

[7] It is questionable to use a depth of closure of 10 m for very long-term runs (centennial or millennia time scales). This concept was designed to be used at yearly scales and, when time scale increases, it has been observed that it usually increases (e.g. Hinton & Nicholls, 1998). In the study area, non-published data show that beach profiles along the northern part of the Ebro delta taken 20 years later than the work of Jiménez and Sánchez-Arcilla (1993) presented significant bottom changes at locations deeper than 10 m. Moreover, if authors used the inner shelf bathymetry to identify the extension of ancient lobes (following Canicio and Ibañez, 1999), changes are observed down to 20 m water depth. If depth of closure is increased, deltaic plain growth rates will be smaller for given sediment supply and wave scenarios.

We realize the shortcomings of a 1-line shoreline model applied over ~1000 year timescales, see also our response to reviewer #1. The increase of the shoreface depth over time becomes intertwined with the simultaneous translation (erosion and progradation) of the shoreline such that an accurate measure of the depth of closure across the entire shoreface and throughout all 2000 years would be practically impossible to constrain. Although we cannot model all scenarios, we have added discussion of closure depth, including the time component, to the manuscript.

Our model uses a shelf-slope such that the Ebro progrades into deeper waters over time. Erosion actually only occurs up to a shoreface depth (see also Ashton and Murray, 2006, part 1). We now state this more clearly. [8] How well wave conditions are correlated with NAO? Authors only mention how transport rates change with NAO positive and negative phases but not how significant (in statistical sense) variations are. This is important to support the hypothesis of no significant change in wave conditions during the period of simulation.

We are not sure we fully understand the reviewer's question. However, to make our analysis more straightforward, we have altered the presentation of the data in Fig. 9b. It now shows more clearly the modern potential littoral transport across all wave angles (a function of only wave height, wave period, and wave directions) as a function of the modern NAO index, without the binning.

Additionally, we do not wish to imply the NAO does not affect the wave climate, but we state that we do not find evidence of a significant effect of the NAO on the annual average potential sediment transport Qs,max. We now make this point more clear in section 4.3.

[revised manuscript text omitted]
 for across longer timescales studies, the shoreface depth should be-increasesd as a result of to take into account the occurrence of lower frequency (storm) events (Hands, 1983). As an potential -indicatorion of this, Guillén and Palanques (1997b) found that the sand-mud transition of the Ebro delta is located at approximately 12 m water depth based on bed-surface samples. In our centennial time-scale modeling of the Ebro delta we take advantage of a recent quantitative analysis of shoreface depthsevolution (Ortiz and Ashton, 2016), which suggests morphological response rates may set the effective shoreface closure depth. that fFor 1 m wave heights, a 100-yr depth

35 depth of 10 m.

Based on calibration studies of Jiménez and Sánchez Areilla (1993) we use an adjusted CERC constant of 0.12 m1/2s-1-compared to a more typical 0.18 m1/2s-1-(Komar, 1998).-The characteristic shoreface slope (0.01) and shelf slope (0.002) are set based on the

of closure is approximately 40% deeper than a 10-yr timescale depth of closure. In our model, we therefore choose a shoreface

geometry offshore of the of the Ebro Delta (Guillén and Jiménez, 1995; Jiménez and Sánchez-Arcilla, 1993). The inclusion of a shelf slope in CEM makes the delta plain prograde slower as it builds out into deeper water further from the coast (Ashton and Murray, 2006). DeltaShoreline retreat maintains a minimum shoreface depth of 10 m. This approach results in a more realistic mass balance, yet does not fully While this behavior is obviously a simplification of the nacapture potential long-term shoreface

- 5 tural-dynamics; the latter would be difficult without -of the Ebro-Delta, the absence of appropriate data to further constrain centennial--scale measurements of shoreface dynamics<del>shoreline change would make the application of more detailed models</del> ineffectivedubious.

[revised manuscript text omitted]

Secondly, the sediment transport patterns along the spits can be cast in the framework proposed by Ashton et al. (2016). Along the barrier sections of the Ebro DeltaEbro delta spits, the computed alongshore sediment transport gradients are nearly zero, whereas measured shoreline retreat is approximately 10 m yr-1 (Fig. 4c). This suggests that in these regions overwash is driving coastline

and is generally erosional up to a fulcrum point, where alongshore sediment transport is maximized and erosion transitions into deposition (Ashton et al., 2016). The measured and predicted shoreline change indicate that the northern and the southern spit are indeed depositional and are prograding at approximately 10 m yr-1 (Fig. 4c).

**3.5 **River Profile Modelmodeling**

- 5 We investigate the response timescales of the river basin to climate and land-use changes using an exploratory 1-D river profile model (Parker, 2004). In this model, sediment is not merely a passive tracer, but interacts with the bed elevation to reach a longitudinal profile in morphodynamic equilibrium (Carling and Cao, 2002). The interaction between flow and topography creates a dynamic model – rivers are not treated as static pipes – which allows us to use the computed longitudinal profiles together with the observed modern longitudinal profile to investigate potential past and present sediment transport conditions. Additionally, by
- 10 focusing on the interaction of the flow with the channel bed, we can model the bed material load – the sediment that makes up most of the delta shoreface (Maldonado, 1975) – while we ignore the finer grained material-washload that is mostly deposited farther offshore. In the absence of subsidence or sea level changes, and if the bed material load and the flow discharge are in equilibrium, the bed slope does not change and the capacity is equal to the supply.
- 15 The channel bed in the model is freely erodible and our approach is therefore strictly applicable to alluvial, transport-limited systems (Parker, 2004). A similar 1-D river profile model was recently applied to study timescales of sediment supply decreases in the Mississippi River (Nittrouer and Viparelli, 2014). Their study suggested a long (O 100 yr) delay between dam construction ~1000 km upstream and sand load changes near the coast.
- 20 The 1-D river profile model assumes normal flow conditions, such that a width-averaged momentum balance connects bed slope and flow depth to bed shear stress. Flow depth in the channel is determined using a Manning-Strickler formulation for the flow resistance (Parker, 2004). The model uses Because of the gravel bed of the Ebro River (Vericat and Batalla, 2006) we use the Meyer-Peter and Muller (1948) equation to calculate fluvial bed-material loadsediment transport (kg s-1),

$$Q_{r} = I \rho_{s} B \sqrt{RgD} D \alpha_{t} \left\{ \left( \frac{Q^{2} k_{c}^{1/3}}{\alpha_{r}^{2} g B^{2}} \right)^{3/10} \frac{S^{7/10}}{RD} - \tau_{c}^{*} \right\}^{2.5},$$
(3)

- 25 where R is the submerged specific gravity of the sediment (1.65); g is gravity (m s-2); D is the median grain size (m) which we choose to be the littoral grain size of 0.2 mm (Jiménez and Sánchez Arcilla, 1993);  $\alpha_{1}$  and  $\alpha_{7}$ ,  $\alpha_{2}$ , and  $n_{4}$  are flow and sediment transport coefficients;  $Q_{flood}$  is the flood a representative flood discharge (m3 s-1);  $k_c$  is the bed roughness (m); S is the channel bed slope; *I* is the flood discharge intermittency;  $\rho_s$  is the sediment density (kg m-3); *B* is the channel width (m);  $\tau_c^*$  is a critical Shields stress for sediment motion; and  $\tau_{e}$  is the non dimensional critical bed shear stress for sediment motion (0.0495) (Parker, 2004). 30 See table 1 for an overview of the model parameters.

From equation (3) we can observe that the intermittency I, the flood discharge Q, and the grain size D are sensitive parameters for the fluvial sediment load estimates. The flood intermittency factor I characterizes fraction of time (generally a year) the river is in flood. Frequently, this The first two are flooding characteristics. Often, the flood intermittency factor *L* is rescaled with a particular

flood discharge to match an observed annual fluvial sediment flux  $Q_r$  (Wright and Parker, 2005). However, the pre-dam fluvial 35 sediment flux of the Ebro River is poorly constrained, so here instead we estimate the flood intermittency I directly from flow records from Batalla et al. (2004) to generate an, but this, does not result in an independent estimate of the fluvial sediment flux. To estimate flood intermittency, we first fit a function to the Because of the poorly constrained pre dam fluvial sediment flux of the Ebro River, here instead we choose to estimate the pre dam intermittency using the flow-exceedance frequency tablesstatistics of Batalla et al. (2004), The first two are flooding characteristics. Often, the flood intermittency factor *I* is rescaled to match an observed fluvial sediment flux  $Q_r$  (Wright and Parker, 2005). A discharge of approximately about 900 m3s-1-represents the critical flood discharge to move bedload, (Vericat and Batalla, 2006)which occurred oftenpre dam - Commonly, the flood intermittency factor *I* is selected to match an observed fluvial sediment flux  $Q_r$  (Wright and Parker, 2005). To convert flow exceedance to floodintermittency, we first fit a trend to the flow exceedance frequency data,

5

20

$$Q(e) = 550 \cdot e^{-0.25}, \tag{4}$$

where *Q* is Ebro River discharge (m3 s-1) and *e* is the pre-dam exceedance frequency (i.e., e = 0.1 indicates a discharge that is exceeded 36.5 days each year). From equation (4), a representative flood intermittency for an annual bed-material load  $Q_r$  can be estimated by Because all the flow above a threshold for motion contributes to the annual bed material load  $Q_{r_2}$  a representative flood intermittency should-takinge into account all the exceedance frequencies floods from an extreme flood (e = 0) to a critical exceedance frequency for bed-material-load motion (700 m3 s-1, or  $e_{crit} \approx 0.25$ ) (Vericat and Batalla, 2006). Furthermore Because1

sediment transport is non-linearly related to flow, so we do not integrate equation (4) directly, but rather we convert it to a sediment 15 flux proxy based on discharge ( $Q^2$ ). Formalized, the flood intermittency of a particular flood magnitude can be described as,

$$I(e) = \frac{1}{Q(e)^2} \int_{0}^{e_{crit}} Q(\varepsilon)^2 d\varepsilon .$$
(5)

In the river profile model we choose a flood discharge of 900 m3s-1 which occurred relatively often with a pre-dam exceedance frequency *e* of 0.15% (Vericat and Batalla, 2006). For a 900 m3s-1 flood, equation (5) evaluates to an intermittency *I* of approximately 0.3. In other words, the instantaneous bed-material load of a 900 m3s-1 flow roughly corresponds to an annual average-fluvial bed-material load if we use a 30% *intermittency* factor-flow, which we therefore use in the model.

The thirdfinalthird sensitive The normal flow assumption is invalid in the backwater zone near the delta, where the channel aggrades and progrades (Hotchkiss and Parker, 1991). Technically therefore, the apex of the delta should be considered the downstream boundary of the fluvial profile model. However, as Chatanantavet et al. (2012) recently showed, annual flooding eycles in the backwater zone often create a condition where aggradation during low flow is nearly balanced by erosion during high flow. This (near) balance suggests that in terms of bedload volumes, delta progradation is significantly larger than channel aggradation and, therefore, that the absence of a backwater zone in our normal flow model only results in a limited underestimation of the fluvial sediment supply to the river mouth when considering centennial timescales.parameter affecting in the fluvial profile model is the grain size. The Ebro River is a gravel-bed river (most mobile  $D_{50}$  is ~ 10 mm) (Vericat and Batalla, 2006), so

- 30 aggradation and erosion due to divergences in the bed-material load should be modeled using gravel size sediments. However, the median sediment size of the Ebro shoreface is sand (~0.2 mm) (Jiménez and Sánchez-Arcilla, 1993). In the coupled fluvial-delta system we should therefore consider the sand load as the representative bed material load volume at the river mouth. To retain the simplicity of a unimodal fluvial profile model we choose a 10 mm median bed-material load grain size to compute the timescales of profile incision and aggradation. Furthermore, g Given the relatively constant slope of the Ebro River (S = 5.8·10-4F, Fig. 5), we
- 35 assume that the bed material load at the Ebro delta should be roughly similar to the bed-material load further upstream despite the change in the median grain size.

Applying the model based on the pre-dam fluvial and discharge conditions, the median bed-material transportload grain size, and the observed slope-up to 450 km upstream of the delta,  $(D_{50} = 10 \text{ mm}, Q_{flood} = 900 \text{ m}^3 \text{ s}^{-1}, \text{ I} = 30\%, \text{ S} = 5.8 \cdot 10^{-4})$  we find an annual average bed-material load transport rate  $Q_r$  of 70 kg s-1 (2.2 MT yr-1). This estimate however is sensitive to the bed roughness  $(k_c)$ which we estimate at 100 mm, ~3 times the bed material  $D_{50}$  (Vericat et al., 2006).

5

10

**to We model the Ebro drainage basin as a single channel representing an average of its tributaries.**

Following Jiménez and Sánchez Arcilla (1993), we choose a grain size of 0.2 mm for the fluvial profile model. This grain size is mostly transported during floods of 900 m3 s-1 or larger; flows which during pre dam conditions were exceeded 15% of the time (Batalla et al., 2004). The fluvial profile model uses one flood magnitude with an intermittency factor rather than an exceedance

- frequency. We have estimated an intermittency factor by fitting and integrating a logarithmic trend to the flood frequency analysis data of Batalla et al. (2004). This integration shows that the sediment load of a 900 m3s-1, 15% *exceedance* frequency flow roughly corresponds to a 900 m3s-1, 30% *intermittency* factor flow, which we therefore use in the model.
- We can compare the predictions from the model to the observed modern river profile and see how close the modern profile is to equilibrium. The modern Ebro River profile (Fig. 5) shows an approximately constant slope up to the confluence with the Arga River, 450 km upstream. Applying the model based on the pre-dam fluvial and discharge conditions (D50=0.2 mm, Qflood=900 m3 s-1, I = 30%, Qr = 70 kg s-1), we find that the equilibrium slope is estimated surprisingly well (5.8-110-4, Fig. 5b). Note that the observed channel slope remains constant upstream of the confluence with Cinca River even though the flood discharge decreases significantly. This could be due to different channel bed grain sizes between the Cinca River and the Ebro River upstream of this confluence. We model the Ebro drainage basin as a single channel representing an average of its tributaries. A spatially explicit
  - model of the Ebro basin would be a significant departure from our exploratory model approach.
- Thise 1-D river profile model requires the choice of an upstream boundary, representing the average location of the fluvial
  discharge and sediment supply into in-the drainage basin. The choice of an upstream boundary is important because it acts as a first-order control on fluvial sediment transport timescales from the drainage basin to the delta. To find an appropriate upstream boundary, we calculated the pre-dam morphologic (2-year) flood discharge along the Ebro-riverEbro River relative to the discharge at the delta from existing hydrologic records (Batalla et al., 2004). We set the upstream boundary condition at 450 km upstream of the delta, where the Ebro-riverEbro River pre-dam morphologic (2-year) flood discharge is 50% of its final discharge at the delta
  and a clear discontinuity in the longitudinal profile occurs (Fig. 5). Note that the observed channel slope remains constant upstream
- of the confluence with the Cinca River even though the flood discharge decreases significantly, a sign of fluvial or sedimentological heterogeneity within the drainage basin. However, a spatially explicit model of the Ebro basin taking into account these heterogeneities would be a significant departure from our exploratory model approach. Applying the model based on the pre-dam fluvial and discharge conditions and the observed slope up to 450 km upstream of the delta,  $(D_{50}=10 \text{ mm}, Q_{floor}=900 \text{ m}^2 \text{ s}^4, \text{I}=$  $35 \quad \frac{30\%, \text{S} = 5.8^{\circ}10^{-4}}{10^{\circ}}$  
[revised manuscript text omitted]
           | Туре  | Lat  | Lon | Water        | Wave         | Wave       | Qs,max                | $\mathbf{R}^2$ | Data        | Reference                |
|----------------|-------|------|-----|--------------|--------------|------------|-----------------------|----------------|-------------|--------------------------|
|                |       | °N   | °E  | depth        | height       | period     | (kg s -1 ) |                | period (yr) |                          |
|                |       |      |     | ( m ) | ( m ) | (s) |                       |                |             |                          |
| Cap Tortosa    | buoy  | 40.7 | 1.0 | 60           | 0.8          | 4.1        | 47.9                  | 0.89           | 1990-2011   | Bolanos et al., 2009     |
| Tarragona      | buoy  | 41.0 | 1.2 | 24           | 1.0          | 5.5        | 72.4                  | 0.86           | 2004-2011   | Puertos del Estado, 2015 |
| MedAtlas       | model | 40.0 | 1.0 | 222          | 0.7          | 4.0        | 48.3                  | 0.76           | 1992-2002   | Gaillard et al., 2004    |
| Hipocas        | model | 40.8 | 1.0 | 68           | 1.1          | 4.9        | 71.1                  | 0.87           | 1958-2001   | Sotillo et al., 2005     |
| Wavewatch III® | model | 40.8 | 0.8 | 63           | 0.7          | 4.9        | 31.1                  | 0.86           | 1979-2009   | Chawla et al., 2013      |

Table 32. Overview of the four river profile model experiments and their final equilibrium slope and bed level change. Q is the fluvial flood discharge,  $Q_r$  is the upstream fluvial sediment supply, i is the initial antecedent fluvial environment, and f is the final fluvial environment.

| Description                                          | $Q_i (m^3 s^{-1})$         | $\mathbf{Q}_f(\mathbf{m}^3\mathbf{s}^{-1})$ | $\mathbf{Q}_{r,i}$    | $\mathbf{Q}_{r,f}$    | Slope (i)                | Slope (f)     | Upstream bed     |
|------------------------------------------------------|----------------------------|---------------------------------------------|-----------------------|-----------------------|--------------------------|---------------|------------------|
|                                                      |                            |                                             | (kg s -1 ) | (kg s -1 ) | (·10 -4 )     | (·10-4)       | level change (m) |
| Sediment x2                                          | 900                        | 900                                         | 35                    | 70                    | <del>2.94.0</del> | 5.8           | 80 130    |
| Discharge x2 1.5                              | 600 4 <del>20</del> | 900                                         | 35                    | 35                    | 5.8                      | 2.94.0 | - 130 80  |
| Discharge $x1.5$ and $\&$ sediment x2                | 4 20600             | 900                                         | 35                    | 70                    | 5.8                      | 5.8           | 0                |
| Discharge $\underline{x1.5 \& and}$ sediment $x2 w/$ | 4 20 600            | 900                                         | 35                    | 70                    | 5.8                      | 5.8           | 0                |
| delay                                                |                            |                                             |                       |                       |                          |               |                  |